



# Modelling the influence of light on the biological characteristics of coastal waters

Paulo F. Lagos[1], Amandine Sabadel[2], and Miles Lamare[1]

[1]Department of Marine Science, University of Otago, 310 Castle Street, Dunedin, New Zealand
[2]Department of Zoology, University of Otago, 310 Castle Street, Dunedin, New Zealand
**Correspondence:** Paulo F. Lagos (pau.lagosj@gmail.com)

**Abstract.** Light is an important regulator of photo-chemical and photo-biological processes in coastal areas. However, understanding how the atmosphere-ocean interaction drives changes in the amount of light entering coastal waters and how changes in the underwater light environment influence the biological characteristic of coastal water can be challenging due to the complex oceanographic dynamics of these areas. Here, we empirically describe the seasonal relationships between meteorological and oceanographic variables over a three year period and quantify the effect light have on the productivity of a coastal area off the Otago coast, New Zealand, through the application of an oceanographic-biological model. The model quantifies changes in the production-biomass ratio (PP/B) (i.e. rate of production of organic matter from phytoplankton produced per unit of total organic biomass) using measurements of the underwater attenuation coefficient, particulate organic carbon, chlorophyll-a and sea temperature. The sensitivity of the model to input data was estimated by comparing the PP/B ratio predicted from Chl a concentrations derived from field measurements of the attenuation coefficients of PAR light $K_d(m^{-1})$ and Chl a concentrations derived from remote sensing data of $K_d(m^{-1})$. The results presented here indicate a mild increment in solar radiation partially driven by increased wind speeds and reduction of cloud cover, ultimately producing small increments in the amount of solar radiation penetrating the water column, especially during summer. The model formulated, predict important seasonal shifts in the PP/B ratio. These shifts are driven by the rate at which light decays and likely modulated by the frequency of wind speeds that favour increments of the thermoclines depth and an increment of sea surface temperatures in the area.

## 1 Introduction

Modelled forecast for this century shows that light and other environmental variables will undergo significant changes, likely affecting the underwater light environment (Liley, 2009; Zepp et al., 2011; Shears and Bowen, 2017). In the open ocean, solar radiation is the most important factor forcing atmospheric and ocean circulation and can typically produce changes at the biological level, including shifts in primary production. However, in coastal areas shifts in productivity largely depend on the natural seasonality patterns and on the complexity of the coastal system (Ma et al., 2019; Gao et al., 2019). This complexity is partially driven by changes in the concentration of both dissolved and particulate, inorganic and organic matter, and changes on primary productivity i.e. Chlorophyll a (Chl a) and can strongly influence the depth at which light penetrates in coastal waters. Moreover, the interactions with several meteorological variables, including temperature, wind speed, cloud cover and



precipitation patterns (EEAP, 2019) can make it challenging to predict shifts in productivity, specially considering the rates of
change observed in coastal ecosystems due to climate change (Bell et al., 2017).

Current climate change projections for the Otago region for the period 2013–2050 indicate a likely increase in temperature
of 0.6 – 0.9ºC by 2040, and of 0.6 – 2.8ºC by 2090 with an increase in wind speed of 2 to 5 knots (Bell et al., 2017; Law
et al., 2018). These climatic trends have the potential to produce changes in the amount of solar radiation penetrating the
water column; however, the magnitude at which temperature and wind speed or other environmental variables influence the
penetration of light or its effects on the overall productivity of coastal area is unknown around the Otago coast, New Zealand.

Changes in the amount of light reaching specific depths of the water column can be captured by the diffuse attenuation
coefficients of downwelling irradiance, $K_d(\lambda)$, defined as the rate of decay of downwelling spectral radiation for a given
wavelength $\lambda(280 - 490\,nm)$ with depth (Cao et al., 2014). Direct field measurements of attenuation coefficients (Tadetti and
Sempéré, 2006) and, more recently, high-resolution remote sensing data obtained from satellites, have been used to assess
the impacts of chromophoric dissolved organic matter, primary productivity, cloud cover, and temperature on incident solar
radiation reaching the surface of the earth or specific depths in the ocean (Ahmad et al., 2003; de Lange et al., 2003; Johannessen
et al., 2003; Lindfors and Arola, 2008; Xiong et al., 2020; Cao et al., 2014). These measurements have also allowed the
modelling of light penetration through the water column (Taylor et al., 1997; Kim et al., 2015; Bowman et al., 2018). Current
satellite open-access products, however, do not include measurements of short ultraviolet wavelength, and the use of open-
access remote sensing products is only possible in the visible band spectrum. For this reason, the behaviour of UV wavelengths
in the ocean has been, in most cases, interpolated and interpreted from direct reflectance products through the implementation
of complex models. (Mobley, 2001; Pan and Zimmerman, 2010; Li et al., 2018).

The main aims of this study are to evaluate the effects of light on the biological characteristics of a coastal area and describe
the seasonal atmosphere-ocean connection over a two year period. To achieve this, we use model predictions of the production
to biomass ratio (PP/B) obtained using remote sensing data or field obtained data of $K_d$ as model inputs, using a simple
mathematical model framework that applied data of the attenuation coefficient of PAR light, and the absorption and scattering
coefficients of ocean water to predict Chl a concentrations which, in turn, are used to estimate the primary production/biomass
ratio (PP/B). For this, we re-purpose a model based on the values of absorption $a_W(\lambda)$ and scattering $b_W(\lambda)$ coefficients of
pure sea water to calculate the water decay constant, $K_W$. The water decay constant is, in turn, used to calculate a theoretical
attenuation $K_{bio}$. Here, we used $K_{bio}$ as a secondary attenuation coefficient parameter, that is a function of in situ measurements
of $K_d$ and $K_W$. $K_{bio}$ is then used to predict Chl a values based on a dimensionless sea water specific derived function $(X)$
and the coefficient $(e)$. Once Chl a values were known, the model is used to predict the PP/B ratio as a function of water
temperature, nutrient loads and changes in underwater irradiance with depth. The approach followed in this study complements
similar studies that characterize the attenuation coefficient of optically complex waters using remote sensing data or modelling
approaches (Modenutti et al., 2001; Cao et al., 2014; Mishra et al., 2005; Cao et al., 2014; Giddings et al, 2021) and is
intended as a method to identify the influence of light on productivity either by using open-access remote sensing products or
in-situ data of the attenuation coefficient. In summary, here we (1) describe the physical characteristics of a coastal area of the
Otago Peninsula and study the connection between meteorological and oceanographic data to (2) understand how changes in





$K_d(\lambda)$ influence the biological characteristics of the water column. Finally, we compared model predictions from two model approaches (3) to evaluate differences in predictions of the PP/B using free access remote sensing data with a spatial resolution of 1.5 km, or direct in situ measurements of $K_d(490)$. Because the model's PP/B ratio predictions ultimately depend on Chl a concentrations and temperature, we expected the behaviour of the predicted PP/B values not to differ in a significant way, regardless of the type of data used. Furthermore, due to the natural seasonal complexity of the study area, we hypothesized that

PP/B ratios increase during summer to twice the values of winter, reducing the amount of light penetration the water column by half the values of winter.

## 2   Materials and Methods

### 2.1   Study site description

The coastal waters off the Otago Peninsula are oceanographically dynamic, dominated by the northward flowing Southland

Current, a mix of superficial Sub-Antarctic waters and Sub-Tropical waters (Murdoch, 1989; Garner, 1961). This water mass can be associated with warmer, higher salinity coastal waters, which are typically separated from offshore waters by the Southland front (Sutton, 2003). This system determines local and seasonal oceanographic characteristics of the waters along the Otago Peninsula, with inshore ocean temperatures 2 ℃ warmer than offshore conditions, and with lower limits of temperature between 10 ℃ in winter and 15 ℃ in summer, and salinities between 34.6 – 34.9 PSU (Jillett, 1969). The region that separates

the Southland current from the coast is considered a neritic zone, an area where the euphotic zone can reach the ocean floor. This neritic zone contains a water mass of varying temperatures as a result of transferring surface heat during summer and convection transfer during winter (Jillett, 1969). Other factors, such as inputs of freshwater run-off from the Clutha River, 96 km south of the Otago Peninsula, reduce coastal salinities values to < 34.6 PSU, especially during the months preceding winter. Additional local ocean features, such as the presence of an eddy, north of the Otago Peninsula, are known to facilitate

the retention and recruitment of planktonic organisms (Murdoch, 1989), increasing the optical complexity of the area.

### 2.2   Data acquisition

Measurements of solar radiation (280 - 700 nm) were undertaken seasonally throughout two years, from December of 2016 to December of 2018, exclusively during the summer (December - February) and winter (June – August). Measurements of underwater solar radiation were made using an underwater spectroradiometer (Licor LI–1800UW) on a grid of 13 stations,

covering and area 3.47 $nm^2$ at the entrance of the Otago Harbour (45° 46.12'S - 170° 43.72'E) (Figure 1). Measurements were obtained at 1 m intervals between the sea surface and 5 m depth, exclusively during clear sky and relatively calm ocean conditions (Beaufort scale 0 - 3). All measurements were taken within 3 hours, between 1100 hrs and 1400 hrs, when the solar





radiation is at its daily maximum, and when the zenith angle is between 24.6 – 38.9° in summer and between 67.1 - 75.0° in winter. For each wavelength, raw measurements from the spectroradiometer were calibrated using the following equation:

$$I(d_z) = \frac{acos \cdot ucos}{I(\lambda)} \tag{1}$$

Where $I(d_z)$ was the calibrated irradiance at different depths expressed in $Wm^{-2}$. The variables $acos$ and $ucos$ were the calibration values provided by the spectroradiometer supplier for each wavelength (Table S1 in the Supplement), and $I(\lambda)$ was the raw measured irradiance values for each wavelength between $300 - 700nm$. The calibrated underwater light data was used to calculate $K_d(\lambda)(m^{-1})$ and the light transmittance, $T(\lambda)(m^{-1})$, for each wavelength between $300 - 700nm$, at each station between the surface and 5m using the following equations:

$$K_d(\lambda) = \frac{1}{z(\lambda_1) - z(\lambda_2)} \cdot LN \frac{I(\lambda_1)}{I(\lambda_2)} \tag{2}$$

$$T(\lambda) = \frac{I(\lambda_1)}{I(\lambda_2)} \cdot 100 \tag{3}$$

Where $I_1(\lambda)$ and $I_2(\lambda)$ corresponded to the irradiance values between surface and 5 m depth $(Wm^{-2})$, and $z_1(\lambda)$ and $z_2(\lambda)$ corresponded to the minimum and maximum depth (m). $K_d(\lambda)$ and $T(\lambda)$ were expressed as $(m^{-1})$ and percentage per meter $(m^{-1})$, respectively. In parallel to underwater light measurements, water column profiles of salinity, temperature and density were captured using a CTD profiler (RBR–XR620; RBR ltd, Canada). This data was used to infer the depths of the thermoclines and pycnoclines using the R package "oce" (R Core Team, 2017) to locate the depth at which differences in the salinity and temperature were larger. This data was used to explore the association between meteorological and oceanographic variables during summer and winter conditions.

### 2.3 Meteorological data

A data set with hourly observations of atmospheric measurements from summer 2016 to summer 2018 was obtained from the weather stations array located at the Department of Physics, University of Otago, Dunedin, New Zealand (45º52'S; 170º31'E) and from the New Zealand meteorological service station also located in Dunedin (45º55'S; 170°11'E) (Table 1). These data-sets were combined and used to study the correlation between the meteorological variables and underwater UVR. The data-set included observations of surface atmospheric temperatures, cloud cover, wind speed and direction and total UVR radiation (Table 1).

For statistical purposes, hourly categorical observations of cloud cover data were converted into a continuous variable by creating a cloud cover index (scale from 1 to 5, $Idx$), where the cloud cover description by the Dunedin branch of the New Zealand meteorological service station (45º55'S;170º11'E) was turned into the following numerical values: no cloud cover became 1; cloud cover described as "Few" (FEW) for one or two eights of the sky cover by clouds, became 2; "scatter" (SCT)



for three or four eighths, became 3; "broken" (BKN) for five to seven eighths, became 4; and "overcast" (OVC) meaning the total of the sky is covered by clouds, became 5. When the sky conditions changed thought the day, the daily average between the values (1 to 5) was used as the index. Hourly observations of wind were transformed into wind pseudo-stress components

$(m^2 s^{-1})$, $Z_u$ and $Z_v$, for the East-West and North-South directions, respectively. Vector components were calculated from the wind direction and wind speeds following the equations:

$$Zu = W \cdot Cos\theta \tag{4}$$

$$Zu = W \cdot Sin\theta \tag{5}$$

where W is the wind speed $(ms^{-1})$, and the corresponding wind direction was expressed in angles. A positive $Z_u (m^2 s^{-1})$ represented winds from the west and a negative $Z_u$ represented winds from the east. Similarly, a positive $Z_v$ value represented winds coming from the south and a negative $Z_v$ value represented winds coming from the north (Ponds and Pickard, 1997).

## 2.4    Remote sensing data

We collected three years of open access remote sensing data (Table 1). Since $K_d(490)$ is the only surface attenuation coefficient

available as free-access remote sensing data, observation of $K_d(490)$ were compared against inferred values of $K_{bio}$ obtained from in-situ observations of $K_d(490)$ to confirm similarity in observations. All remote sensing data was obtained from the national aeronautics and space administration (NASA) ocean colour website (https://oceancolor.gsfc.nasa.gov). The level–3 (derived geophysical variables) binned (1.5 km spatial resolution) data was taken by the MODIS (Moderate Resolution Imaging Spectroradiometer) sensor that orbits the planet at an altitude of 705 km and make observations of the entire Earth's surface

every two days. The data is acquired in 36 spectral bands or groups of wavelengths, including ocean colour, phytoplankton and biogeochemistry (bands 8–16) (NASA Goddard Space Flight Centre, Ocean Ecology Laboratory, Ocean Biology Processing Group, 2014). The obtained data sets were converted to non-binned data using the software SeaDAS (version 7.5.3) and QGIS ("QGIS Development Team, 2019). Monthly and seasonal (summer and winter) averaged values of Chl a, DOM and $K_d(490)$ were calculated for a transect of 22.7 km located across the stations of the present study, extending offshore of the Otago

Peninsula. Data points from the region of interest were extracted, and converted into readable raster images, using the software R (R core team 2013) and the packages (package = "ncdf4", package = "ocedata" and package = "raster").

## 2.5    Data analysis

The general characteristics and trends of the meteorological variables were described by calculating the average and extreme values for each variable (average value ± SD) during the studied seasons, and then the annual cycles for each data were

analysed using time series analysis. We calculated smoothed trends using the R package "openair" and compared time series of atmospheric data by measuring the characteristics of the time series such as: strength of seasonality, strength of trend,





level of non-linearity, skewness, serial auto-correlation, self-similarity, and the periodicity of each time series. In addition, the seasonal and monthly averaged relationship between meteorological and oceanographic variables were analysed using principal component analysis (PCA) by combining remote sensing and all meteorological with the inferred values of the thermocline and pycnoclines.


## 2.6 Estimation of chlorophyll-a from attenuation coefficients

To derive Chl a concentrations from $K_d(\lambda)$ values, two different attenuation coefficients, $K_{bio}$ and $K_W(\lambda)$ were used following a well established approach to characterize optically complex waters (Morel and Maritorena, 2001; Morel et al., 2007; Giddings et al, 2021) (Figure 2). $K_{bio}$, includes the contribution of all biogenic components of the water column, its calculation was

approximated by using an $\chi(\lambda)$ coefficient that changes with wavelength and the theoretical exponent $e(\lambda)$. Both, statistically derived functions described by Mobley (1994) and first used on a bio-optical model given by Morel (1988) to describe optically complex waters with Chl a values $< 30\,mg\,m^{-3}$. The approach described $\chi(\lambda)$ and $e(\lambda)$ function values from 400 to 800 nm. Thus, values for UV-wavelengths $< 400nm$ were obtained by fitting a third order polynomial model to the regression between $\chi(\lambda)$ and $e(\lambda)$ (R core team, 2019)(Table B2). Together, equations (6) and (7) were used to estimate Chl a concentration

$(mg\,m^{-3})$ from $K_d(\lambda)$. However, $K_{bio}$ was approximated from in situ measurements of $K_d(490)$ following equation (6) thought its relationship with the water decay constant $K_W$. Theoretically, $K_W$ represents the spectral values of the diffuse attenuation coefficient of pure sea water. $K_W$ was best approximated by its lower values limits, which are expressed by equation (7):

$$K_{bio}(\lambda) = K_d - K_W \tag{6}$$


$$K_W(\lambda) = a_W(\lambda) + (1/2)b_W(\lambda) \tag{7}$$

Where $a_W(\lambda)$ and $b_W(\lambda)$ represented the absorption and molecular scattering for optically pure sea water. The validity of this formulation has been discussed in detail by Smith and Baker (1981), and values used in this study for $K_W$ where those from Smith and Baker (1981) (Table B1). Using equations (6) to (8), the estimation of Chl a from $K_d(\lambda)$ values was obtained

as follow:

$$K_{bio}(\lambda) = K_W(\lambda) = \chi(\lambda)(Chla)^{e(\lambda)} \tag{8}$$

In this study we assume that $K_d(\lambda)$ values change with wavelengths and interpretation of Chl a values were obtained from mathematical derivation of equation (8), proposed by Morel (1988). Finally, values of transmittance, $T(\lambda)$, were used as a secondary tool to explore the extent of the relationship between Chl a and the measured values of $K_d(\lambda)$. This was achieved by

implementing the log-log regression between field obtained $K_d(\lambda)$ and modelled $K_d(\lambda)$ coefficients, with the aim to evaluate if the behaviour of the $T(\lambda)$ values followed the expected decay pattern with depth at different levels of Chl a.





## 2.7 Biological model for PP/B estimation

A modified version of a model previously used to estimate PP/B ratios (Hernández et al., 2012; Taylor et al., 1997) was applied to quantify variations in PP/B due to variations in light. The first models was adjusted using estimated Chl a values derived from remote sensing $K_d(490)$ values and the second model adjusted using Chl a derived from field obtained of values $K_d(490)$ between the ocean surface and 5mt depth. Both models, however, used POC $(mg\,C\,m^{-3})$ data from satellite observations to calculate the effect of light on the PP/B ratio following the formulation listed in equations (9) to (11).

$$PP/B = \frac{Vmt + \alpha + I_Z}{\sqrt{Vmt^2 + \alpha^2 + I_Z^2}} \cdot \frac{[N]}{K_n + [N]} \cdot C : Chl\,a \tag{9}$$

$$Vmt = a \cdot b^T \tag{10}$$

$$[N] = POC \cdot \frac{\left(\frac{6}{5}\right)}{b} \tag{11}$$

With PP being primary production $(mg\,C\,m^{-3})$ and B the phytoplankton stock $(mg\,\text{Chl a}\,m^{-3})$. The terms on the right of the equation represented the light and nutrient limitation: with $VmT$ the temperature dependant maximum growth rate function at light saturation, calculated using equation (10); where $\alpha$ is the initial slope of the production/irradiance curve; $I_Z(W\,m^{-2})$ at depth z(m); [N] the nitrogen concentration for New Zealand coastal areas, calculated converting POC data using equation (11); $K_n$ the half saturation constant for nutrient uptake; and with C:Chl a limited to values $< 20$ (Taylor et al., 1997) (Table2). While the C:Chl a ratio depends on light and nutrient limitation, here we fixed it to values from 0.003 to 0.01 $(mg\,\text{Chl–a}\,(mg\,C)^{-1})$ in the two models, to level the amount of input variables between the two models and also due the lack of data for the study area. Therefore, the fix values were based on derived values of Chl a from remote sensing and field obtained data and values reported by other authors (Cloern et al., 1995).

The values form the dimensionless function $VmT$ were calculated following equation (10), where "t" was the in situ temperature taken from the CTD profiles, "a" was the maximum growth rate at 0 °C set to 0.8 $d^{-1}$, and "b" an specific phytoplankton growth parameter set to 1.06 following (Oschilies and Garçon, 1999; Koné et al., 2005).

Because the implementation of the model followed a less complex approach, which its primary focus was to assess the capability to track changes in the PP/B ratio in time, here values of [N] were assumed constant with depth and the calculations of the nutrient concentration were made using surface POC values obtained from remote sensing measurements. The transformation assumed a C:N ratio of (5:6) according to equation (11), where "b" is the molar weight of N.

## 3 Results

### 3.1 Seasonal changes in meteorologic-ocean conditions

Maximum surface levels of solar radiation between 11:00 – 14:00hrs during the austral summer were between two to three times higher than values during the austral winter when few to no clouds were present. During summer, solar radiation and





cloud cover displayed larger daily temporal variability. However, the periods of peak solar radiations during winter were longer compared to summer months (Table 3). For instance, during summer, only 26.5% of the time the sky presented NCD conditions, while the remaining 74% of the time clouds conditions varied from FEW to OVC conditions. As a result, cloud cover produced a significant reduction of the incident solar radiation levels (two-way ANOVA; $F_{(4,2379)} = 19.78; p < 0.05$).

With OVC conditions decreasing the mean intensity of solar radiation between 34 to 37%. While, SCT produced a reduction 21.3% and BKN cloud conditions produced an overall reduction of 34% in solar radiation reaching the surface of the ocean. In contrast, during winter months, all cloud conditions had significant effects on the total mean values of solar radiation, with OVC conditions reducing solar radiation levels between 55 to 66.4%, an average reduction 1.5 times higher than other clouds conditions (Table 3).

Average atmospheric temperature during the three year period were $10 \pm 4.7$ ℃ higher in summer than in winter and higher temperature were positively associated with an increment of the meridional wind velocity, that pushed important reductions in the amount of clouds during summer months, time of the year when total wind speeds were stronger (summer = $7.68 \pm 5.04$ kt and winter = $5.15 \pm 4.46$ kt). Time series analysis of solar radiation showed a small increment of solar radiation during summer months with a mild seasonal component (Table A1), while cloud cover time series show a mild decrease of clouds

cover in time (Fig.A1). The Lyapanuv exponent for both time series, which dictated the rate of separation, was similar between solar radiation ($\sim$0.56) and cloud index ($\sim$0.51) time series (Table A1). This suggested the existence of a similar behaviour between cloud cover and solar radiation (Fig.A1).

The oceanographic conditions for the study area followed a trend similar to the atmospherical data, with ocean water temperature higher in summer compared to winter values and salinity values ranging from 34.2–34.6 PSU in summer and 33–34.9

PSU in winter (Table 3). Observations of remote sensing data and field data showed an increment in time of ocean temperatures (3). However, observations of $k_d(490)$ and remote sensing and observations of $k_d(320)$ showed opposite trends (Table 3) but a similar coast-offshore gradient with overall higher values of $K_d$ closer to the coast (Fig.C5). This gradient was most evident at the beginning and the end of the summer months but was less apparent during winter, when values of Chl a across the study area remained relatively unchanged (Fig.C3).

Results from PCA analysis showed that although values of POC were significatively different during summer compared to winter months (ANOVA, $F_{(1,58)} = 4.401; p = 0.04$)(Table 3), with concentration generally decreasing with distance from the coast (Fig.C2), the main drivers of differences in the characteristic of the water column are seasonal (Fig.4a). With higher Chl a and $K_d(490)$ driving most of the variation in summer (Fig.4b) and higher $k_d(320)$ in combination with a shallower halocline driving most of the variation in winter (Fig.4c). For instance, significant differences in mean $K_d$ values between

summer and winter months (ANOVA, $F_{(1,58)} = 4.401; p = 0.04$) were found with lower values during winter compared to summer (Table 3). Therefore, more light is able to reach deeper layers of the water column during winter months. Full analysis of the $K_d$ from $300-700$ nm showed statistical differences between values of $K_d$ at surface and values of $K_d$ at 5m depth for all stations (ANOVA, $F_{(11,60)} = 2859; p \leq 0.05$) (Fig. 3) and seasonal differences in the $K_d(300-700)$ between summer and winter (ANOVA, $F_{(1,75)} = 134.1; p \leq 0.05$).





### 3.2 Relationship between spectral values of modelled $K_d$, field $K_d$ and field T

The specific absorption $(a_W)$ and scattering $(b_W)$ coefficients for pure sea water proposed by Smith and Baker (1981) used in this study followed a rapid exponential decay, with a rapid decrease of $a_W$ with increasing values of $b_W$ (Table B1) and differed from the coefficient of pure water first proposed by Pope and Fry (1997), which follow a logistic behaviour.

Knowing the behaviour of these coefficients allowed for accurate parameterization of $K_W(\lambda)$. $K_W(\lambda)$ values calculated in this study showed an increase with wavelength and differed substantially at wavelengths above 600 nm (Fig. B1). The water decay constant allowed for the calculation of $K_{bio}$ that included the contribution of all the biogenic components of the sea water, following equations (6) to (8). The behaviour of this theoretical attenuation coefficient was similar to field measured $K_d$, despite the modelled $K_{bio}$ and field measured $K_d$ obtained from two independent methods. Overall, a positive significant correlation between $K_d$ and $K_{bio}$ was found ($R^2 = 0.81$, p ≤ 0.05), which indicated a strong wavelength dependency of the variations of the field obtained $K_d$ rather than observed changes caused by environmental factors (Fig.B2). Furthermore, as expected field obtained measurements of transmittance decayed in a logistic way with increased modelled and field obtained $K_d$ values (Fig.5a and Fig.5b).

### 3.3 Derivation of chlorophyll a

Calculated values of Chl a based on the wavelength dependent coefficient $\chi(\lambda)$ and exponent $e$, indicated similar concentrations of Chl a during winter $(1.10 \pm 0.23 \, mg \, m^{-3})$ and summer $(1.60 \pm 0.53 \, mg \, m^{-3})$. However, Chl a obtained from remote sensing data were slightly higher both in winter $(1.11 \pm 0.45 \, mg \, m^{-3})$ and summer $(1.65 \pm 0.50 \, mg \, m^{-3})$.

The parameters used for the calculations of Chl a, $a_W$ and $b_W$, were those of light as a function of each wavelength in pure sea water and led to some overestimated values of Chl a. Despite the overestimation, values of Chl a followed a log-log association with the modelled $K_d(\lambda)$ (Fig. C1d and C1e), which indicated a wavelengths dependent response of Chl a to different $K_d$ values, with changes in $K_bio$ at lower wavelengths (UVR 300-380 nm) occurring at a faster rate than at higher wavelengths (PAR 390-700 nm) and when levels of Chl a are $< 10 \, mg \, m^{-3}$ (Fig.5c).

Statistical tests performed on the modelled Chl a, showed significant changes in Chl a between summer and winter (ANOVA, $F_{(1,1218)} = 4.834; p = 0.028$) (Table 3) and between stations (ANOVA, $F_{(11,1208)} = 3.6; p < 0.05$), with values of Chl a higher at the east-ward stations located closer to the coast ($1.25 \, mg \, m^{-3}$), which is a similar pattern to the one found for the summer MODIS-aqua Chl a estimates. However, the model was not able to reproduce statistically significant changes in Chl a associated with depth, as depth correction factors were not included in the formulation of the model and Chl a values were a function of $a_W$ and $b_W$, and measured $K_d$ values rather than more complex dynamics. As changes with depth in Chl a could not be captured by remote sensing, values of Chl a for the MODIS-Aqua-based model were assumed to be constant. Although the calculations of modelled PP/B values integrated a temperature correction function $VmT$ that could utilize values of temperature at different depths, the approach followed using MODIS-Aqua SST to integrate changes in PP/B over time and the $VmT$ function alone did not offer enough variation to capture changes on Chl a with depth.





### 3.4 Field obtained model vs MODIS-Aqua PP/B model.

The first approach taken, using the slope ($\alpha$ parameter) of the lineal model between irradiance and derived Chl a values from remote sensing data and nitrogen derived observations from MODIS-aqua POC, resulted in overestimated values of the PP/B
ratio (284.43 ± 91.5). However, using an $\alpha$ value, specific to New Zealand waters (Bouman et.al., 2018), the model yielded more accurate PP/B ratios (Fig. 6a). PP/B values predicted based on MODIS-aqua data were higher in winter (317.60 ± 81.2) than in summer (258.97 ± 58.2). In contrast, mean PP/B values predicted based on field obtained data were lower, both in summer (111.27 ± 85.0) and winter (292.56 ± 71.2) (Fig.6b). The integration of nitrogen values from POC, assuming no changes with depth, indicated lower values in summer and higher values during winter (Fig.6c), with mean values for the
$VmT$ function oscillating between (0.8 - 1.8) and higher values for the field-based model (1.62 ± 0.185) compared to the MODIS-Aqua-based model (0.87 ± 0.042); (t(5.512) = 9.701, p = 0.0001). However, SST values from MODIS-Aqua followed a similar dynamic to that of the mean seasonal field temperatures obtained from the CTD profiles (Fig.6d). Overall, we found that PP/B ratios from MODIS-Aqua and field obtained models had correlations values above $\approx 45$ when compared with both atmospheric and underwater related variables.

## 4   Conclusion and Discussion

### 4.1   Seasonal effect of atmospheric parameters on solar radiation levels.

The atmospheric components included in this study only represent a sub-set of environmental mechanisms that control incident UVR reaching the surface of the water. For instance, it is known that higher atmosphere aerosols play an important role in reducing incident UVR, while water vapour and seawater aerosols in coastal areas are known to reduce UVR, changing the
reflectivity and enhancing albedo at low near-surface altitudes (Häder et al., 2003; Lovengreen et al., 2005; Thomas et al., 2012; Williamson et al., 2014). Broader-scale modelling of the total solar radiation reaching the ocean is not simple, considering the range and number of variables that come into play, however, studies that incorporate only a fraction of these variables have been successfully and extensively used in the past to study and forecast the impact that solar radiation has on ocean ecosystems (Conde et al., 2000; Ahmad et al., 2003; McKenzie et al., 2008; Lee and Feldstein, 2013). Overall, the results presented here,
confirm that atmospherical variables such as wind speed and air temperature influence solar radiation at a local scale and can be used to explain synoptic variations in the amount of solar radiation reaching the earth surface. We found a positive correlation between UVR and wind speed ($\approx 0.22$), that showed that in the presence of wind speeds above 10 knots, levels of solar radiation increased by a factor of 1.1 in sunny conditions. This response is likely due to the increased movements of clouds or the higher dispersion rate of clouds caused by the wind conditions. Other studies have found similar correlation between UVR
and atmospherical variables ($Z_u$ - UVA = 0.49; $Z_u$ - UVB = 0.43) (Hernández et al., 2012).

Latitudinal small-scale differences in wind speed, atmospheric temperatures and clouds were expected to drive the local differences in incident solar radiation. However, given the naturally occurring fast changes in the atmospheric condition, a characteristic of the area of study, we were also expecting that these trends would be difficult to identify. These differences



could have led to inaccurate conclusions if too many generalizations had been made, which is why a local understanding of
climatological dynamics is considered essential (Williamson et al., 2014). Previous studies have shown that wind speed play
an essential role in altering the concentrations of ozone (Bais et al., 2015), affecting the consistency of the correlation between
wind speed and solar radiation. Here we founds that when wind speed is separated into its wind pseudo-stress components,
the correlation between wind and solar radiation became less clear ($\approx$5). However, based on the wind direction predominance,
when either stronger south-westerlies blew during winter or north-westerlies during summer, levels of solar radiation were
higher, which was consistent with the cloud cover as shown in the results of time series analysis.

We found a high degree of patchiness in cloud cover data due to daily variability, a typical condition of coastal areas in
New Zealand. However, a positive correlation between cloud cover and atmospheric temperatures was observed when different
types of clouds were present. This correlation was strongly dependent on the cloud conditions during the months of summer
and less strong during winter, possibly due to the variable atmospherical conditions present during summer. Unexpectedly, dur-
ing winter, clear sky conditions were more predominant than during summer and the presence of clouds had a more substantial
effect on the levels of solar radiation during winter, reducing values of radiation to more than half under overcast conditions.
During summer this decrease was less apparent, most likely because the higher levels of solar radiation were less affected by
clouds, producing instead physical scattering of the solar radiation which was enough to maintain overall higher levels of solar
radiation regardless of the presence of clouds. These results fit with time series analysis of the atmospherical variables. For
instance, a similar seasonal trend, with closely related seasonality index, was found between UVR and atmospheric tempera-
tures. This contrasted with the results found for the cloud index and wind pseudo-stress time series. It is possible this reflects
the need of more extended time series to foresee higher auto correlation values, and distinguish a more marked seasonal trend
on more variable data due to the natural stochasticity in the dynamic of wind and cloud cover. therefore if more comprehensive
forecasting is required, it is likely that longer time series would be required. Climate change projections for the Otago region
for the period 2013-2050 indicate a most likely increase in temperature and wind speed (Bell et al., 2017). Based on our results,
these changes would increase the amount of solar radiation reaching the first meters of the water column and arguably decrease
the cloud cover. This could promote the need for developing more complex equations, such as used in radiative models first
develop by Mobley during the mid 90's, to understand how light in the atmosphere interact with the ocean.

### 4.2 Atmospheric-Ocean connectivity.

Currents and ocean circulation patterns have an essential function in determining SST off the Otago coast (Jillett, 1969; Mur-
doch, 1989; O'Driscoll and McClatchie, 1989). We found an strong connection between increases in atmospheric and sea
temperatures. However, this connection did not necessarily explain the amount of solar radiation reaching the deeper layers
of the water column, as factors such as nutrient loads and primary productivity seem to play a more pivotal role in altering
$K_d$ values in the region. We found that the atmospheric conditions changed faster during the summer months. These rapidly
changing condition were reflected in the field measurements of ocean temperature and salinity profiles. For instance, evidence
of a strong atmosphere-ocean connection was reflected in the higher variability of the thermocline and pycnocline depths,
which moved deeper during summer, indicating a potentially deeper mixing layer depth, which can partially explain the higher





$K_d(\lambda)$ values found in summer. The Otago coast is influenced by a high discharge of run – off from the Clutha river, bringing more particular organic matter into the study region, potentially driven by an increase of rainfall (Murdoch, 1989). Although, rain data was not included in this study, rainfall on a 6-month average for the Otago region is generally higher in summer (391 mm) compared to winter (367 mm) (Macara, 2015). These differences in rainfall plus higher solar radiation levels driven by ozone movement in the higher atmospheric levels due to wind speed have the potential to increase the productivity at a local scale, thus maintaining higher levels of particulate matter in the first meters of the water column increasing $K_d(\lambda)$ further

Additionally, complementary remote sensing data of Chl a (Fig. C3) and POC (Fig. C2 also followed the changes in SST occurring from inshore to offshore. Moreover, higher surface levels of Chl a and POC were present in the area during summer, which contrast with the lower levels found in winter. Satellite measurements of the $K_d(490)$ also indicated a higher attenuation of the light during summer, which correlated with field-obtained measurement of the same $K_d(490)$. An important observation; however, is that remote sensing data sets used in this study could suffer from small-scale spatial and temporal patchiness. Therefore, for more accurate seasonal representations, it may be necessary to integrate average values over longer periods of time to partially overcome patchiness and added robustness if forecasting models are implemented.

### 4.3 Attenuation coefficients, light transmittance in the Otago coast and modelled attenuation coefficients.

We found differences in $K_d(\lambda)$ inside our study area. For instance, $K_d(490)$ from satellite data was stronger in the areas closer to the coast than in areas further offshore, and a similar pattern was found for in situ measurements of $K_d(320)$. These difference in $K_d(\lambda)$ were easier to visualize using remote sensing techniques, as difference were accentuated as satellites can sample a more extensive area. We found that remote sensing attenuation coefficients $K_d(490)$ correlated with in situ measurements of $K_d(320)$. However, the process of obtaining valid measurements of in situ $K_d(\lambda)$ was extensive and required comprehensive sampling to avoid erroneous measurements. In this study, we obtained $K_d(\lambda)$ values during clear sky and calm sea conditions to minimize the effects of sea state on instrument deployment (i.e. excessive lateral boat movements), which yielded a strong correlation between remote sensing $K_d(490)$ and $T(490)$. Additionally, T($\lambda$) changed inversely proportional to $K_d(\lambda)$ at different wavelengths, and there was a logistic correlation between T and $K_d$ that also changed with depths. At shallower depths (1-2 m depth), there was a steeper change, and minor increments of $K_d$ produced a steep drop in the T values. This was an indication that light, and more specifically short UV wavelengths, are mostly attenuated in the surface layers of the water column. Few studies have previously shown how light behaves in coastal areas of the Otago Peninsula (Lamare et al., 2007), nevertheless considering the oceanographic dynamics proposed by Murdoch (1989); Murdoch et al. (1990) it was expected to observe a linear decay of light with depth and not the abrupt decay in the first meter of the water column as found in the present study. previous studies of phytoplankton and zooplankton in the area suggest high values of primary and secondary productivity in the Otago coast (Jillett, 1969; O'Driscoll and McClatchie, 1989; Takagaki, 2006). These peaks of productivity in the area could explain the higher surface $K_d(\lambda)$ values found. However, to date there are no studies on vertical abundance of phytoplankton or zooplankton in the coastal areas throughout the Otago Peninsula. The most recent study was an M.Sc. thesis that investigated the horizontal variation of the subtropical front (Ramadyan, 2017). In summary, the coastal area off the Otago Peninsula have high $K_d(\lambda)$ values, due to high concentrations of POM, which limits the propagation of high energy





wavelengths into deeper waters. These high concentrations result from the predominant wind conditions that create a shallow thermo- and pycnocline, especially during summer months. Ultimately, retaining POM influx from the Clutha river´s run-off in shallower layers of the water column.

## 4.4 Modelled Chlorophyll-a values and $K_d$ values.

The modelling approach implemented in this study, although based on similar core equations and principles, did not require the use of extensive computational routines, as it can be the case when using other well-established radiative transfer models (Mobley, 2001; Emde et al., 2016). For instance, derived Chl a and modelled $K_d(\lambda)$ values were a function of the absorption and scattering coefficients of pure water, this differed from values previously reported (Barrot, 2006) and change from study to study depending the experimental approach used, which could lead to small differences in results from the model. However, this remains an active area of research in optics and physics. Using these parameters plus others previously described in the methodology section, the model has a consistent behaviour, and modelled parameters have the expected correlations with field measurements. For instance, modelled and field $K_d$ values displayed a positive lineal correlation and wavelength-dependent differences in the modelled quantities. As two different approaches were followed, one calculating a $K_{bio}$, which values depended either on the Chl a concentration or the dimensionless function $\chi$; and the second one, following a theoretical $K_{bio}$ which values were the sum of $K_w$ and empirical measurement's of $K_d$. In both cases, a linear relationship was found between both modelled quantities and field $K_d$ values.

A similar relationship between modelled values of $K_d$ obtained using different parameterizations, and field values of $K_d$ has been described by Kim et al. (2015), who found a correlation between ($\approx 0.47$ to $0.70$) depending on the region of the world, and an overall correlation of ($\approx 0.02$) between Chl a concentrations and $K_d$ values. The Chl a values from this study were derived from a function that incorporates field $K_d$ measurement, the $\chi$ function, and the water decay constant. The derivation of Chl a values from this equation has been empirically proved by Morel and Maritorena (2001) and tested using remote sensing measurements of $K_d$ and SeaWIFS reflectance by Barrot (2006). In both cases Chl a decline with increasing attenuation coefficients and different reflectance ratios. Here, the modelled Chl a values were below the maximum values obtained from satellite data but fell within the range reported for the area by other authors (Murphy et al., 2001; Ramadyan, 2017). The approach followed in this study, from which Chl a were derived from field $K_d$ values, adds to the evidence that within certain boundaries, it is possible to use remote sensing data to study coastal systems. The complexities and biases involved in the use of remote sensing data solely to this purpose has been extensively discussed in the literature (?Pan and Zimmerman, 2010; Cao et al., 2014; Liang et al., 2019). From this perspective, integrative studies that utilise remote sensing data and field measurements to produce a complementary model approach that fill gaps in data might be an appropriate way of dealing with issues regarding patchiness of remote sensing data in coastal areas.

### 4.5 Biological production to biomass (PP/B) model

No data of PP/B has been previously reported for coastal areas around the Otago Peninsula, which made the validation of our model difficult. However, studies including PP/B ratio for other parts of the world, showed that values of the PP/B ratio could





fluctuate between 20 to 400 (mg C mg Chl a-1) depending on the region, which is in line with the results presented here. For

instance, Hernández et al. (2012) reported maximum values of 140 mg C mg Chl a-1 for the central coast of Chile; while Taylor

et al. (1997), following a similar approach as the presented here, reported maximum values close to 300 (mg C mg Chl a-1) and

recognize latitudinal differences in the PP/B rate between 150 to 250 mg C mg Chl a-1 between 35º and 45º latitude. However,

none of these studies discussed coastal ecosystems. The values obtained in this study were within the range reported by other

authors, but it is unknown to which extent general values of the PP/B ratio from coastal marine ecosystems can deviate from

values from open ocean areas or others coastal regions within New Zealand. Herein we calculated two PP/B ratios, one derived

from satellite observations of Chl a and a second one resulting from modelled values of Chl a. In both scenarios, POC data

converted into N concentrations was assumed constant with depth. Both scenarios yielded different results, with satellite-based

PP/B ratios on average higher than ratios obtained from modelled Chl a values. From a mathematical point of view this was

likely to the use of constant values for Chl a concentrations with depth, which leaves the VmT function acting as the mayor

source of variation in the model. Nevertheless, values of the satellite-based PP/B ratio model were similar to those of the Chl

a based PP/B ratios at surface layers of the water column and when values of Chl are lower (Winter months). This means that

if equations that allowed for the expansion in the predictability of Chl a, based on surface measurements, are incorporated in

the future, models with the capabilities to integrate changes with depth in the amount of Chl a using satellite information could

become a feasible tool to study coastal area such as the Otago Peninsula. Finally, the model implemented in this study was

capable of interpreting seasonal changes in the PP/B ratio, in this way it would be interesting to evaluate how a more complex

model that incorporates attenuation coefficients and field measurements of nutrients and Chl a contrast with the findings of the

model proposed on this study.

**Appendix A:  Atmospheric data analysis**

The weather in Otago NZ is highly variable, for this reason we had to first investigate if any weather patterns was observable in

the environmental variables. We chose to use time series analysis over the three year period of hourly observations from 2016

to 2018 using the R package "Openair" that uses a non-parametric method to calculate time series trends using a Generalized

Additive Model (GAM) to find the linearity in the data. Statistical trends are presented in data table (Table S4).

**Appendix B:  Relationship between spectral values of $K_d$, field obtained values $K_d$ and light transmittance.**

We use the specific absorption ($a_w$) and scattering ($b_w$) coefficients for pure sea water proposed by Smith and Baker (1998)

(Table B2). But first, we needed to know the behaviour of these coefficients in order to accurately parameterize $K_w$, whose

values showed an increase with wavelength and differed substantially at wavelengths above 600 nm (Fig. A3) Simultaneously,

when comparing field $Kd(\lambda)$ with modelled $K_d(\lambda)$ values (which is dependent on Kw values), a strong positive lineal corre-

lation was found at all wavelengths (Fig A4) (See Tables B1 to B3, Appendix B for values of ($a_w$) and ($b_w$), and for the values

of the functions $\chi(\lambda)$ and $e(\lambda)$ used to calculate some of the parameters of the model).



**Appendix C: Biological model for derivation of Chl a.**

Because calculation of Chl a values also needed to depend on the biological and physical characteristic of the environment, we choose an approach proposed by Morel and Maritorena (2001), and Morel et al. (2007). In these, $K_w$ values are compared against a theoretical attenuation coefficients $K_{bio}(\lambda)$, whose values incorporate the contribution of biological components from

the water column. When inferring values of Chl a using the coefficient $K_{bio}(\lambda)$, we found that values of Chl a related almost linearly to the values of $K_bio(\lambda)$. Moreover, when comparing $K_bio(\lambda)$ against field obtained $K_d(\lambda)$ values we found a similar linear relationship, though, with a higher degree of scatter as the function $K_bio(\lambda)$, hence creating a higher degree of noise in the correlation (Fig A.5.c). Further, we narrowed the behaviour of $K_bio(\lambda)$ to a specific wavelength (320 nm) and compared it with the values of Chl a, and found an almost linear log-log relationship, indicating that $K_bio(\lambda)$ is capable of reconstructing

the shifts in Chl a levels at higher and lower levels of $K_d(\lambda)$ (Fig A.5.d and Fig.A5.e).

**Appendix D: Modelling of the PP/B ratio.**

With the knowledge of the behaviour of the parameters $K_bio(\lambda)$, $K_w$ and Chl a, the next step was to use these calculated parameters to infer the PP/B ratio, from remote sensing data using the values of the $K_d(490)$ and from field obtained values of $K_d(490)$, in both cases following the equation (11) listed in the main manuscript. For this, first we checked that remote

sensing data followed the expected correlation normally found between ($Chla\ K_d$), which dictates that higher levels of Chl a normally translate into higher values of $K_d$. (Fig. A.5.a and Fig.5.b). Once we establish the correlation followed the expected lineal behaviour, we test the model setting the parameter $\alpha$ to general values between 0.056 to 0.093 following the approaches previously described by Dower and Lucas, (1993) and Aalderink and Jovin (1997). However using these values lead to a slight overestimation of the PP/B ratio (Fig. 7a1. Consequently, given to the sensitivity of the model to the parameter $\alpha$, we set

the parameter to a more parsimonious values of $\alpha$ = 0.83 that represented coastal water within NZ, following the approach described by Bowman et.al., (2018). As a result, the model yielded more accurate values of PP/B ratio as shown in the main article. The increase in the $\alpha$ value pushed the MODIS-Aqua PP/B model estimations higher, most likely because values in the C:Chl a ratio calculate using remote sensing data of Chl a were higher (217.5 ± 174.6) than values of the C:Chl- ratio calculated using in-situ data (205.8 ± 48.5). Furthermore, a positive increase of 0.1 in the $Vmt$ function for the MODIS-Aqua model,

increased the value of the PP/B ratio in the MODIS-Aqua model as terms at both side of equation (11) are multiplicative.



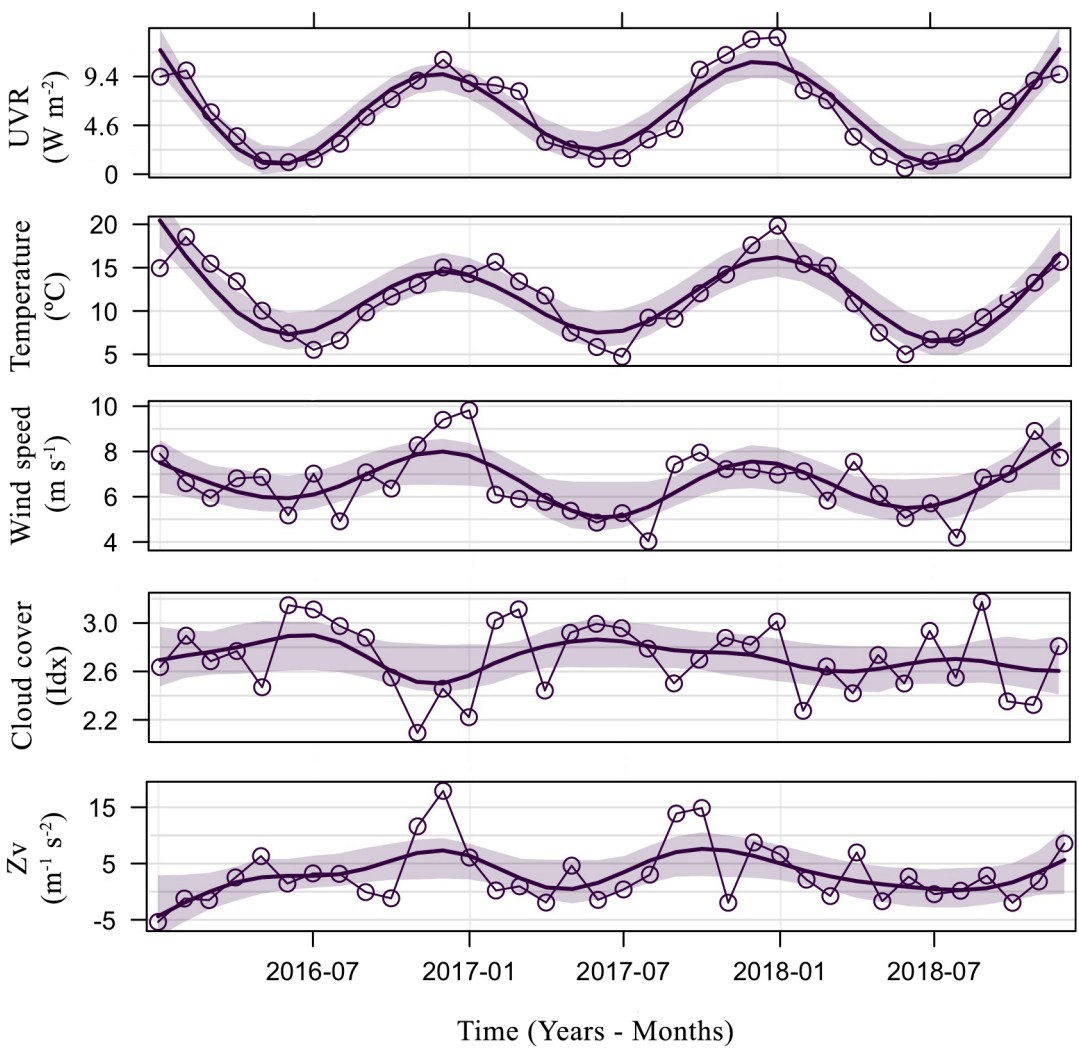

**Figure A1.** Smoothed time series using hourly observations of UVR (W m-2), cloud cover (Idx), atmospheric temperature (ºC), wind speed (m s-1) and latitudinal wind pseudo-stress (Zv) (m-1s-2), from January 2016 to December 2018. Darker lines indicate general trend and shaded areas represent 95% confident interval.




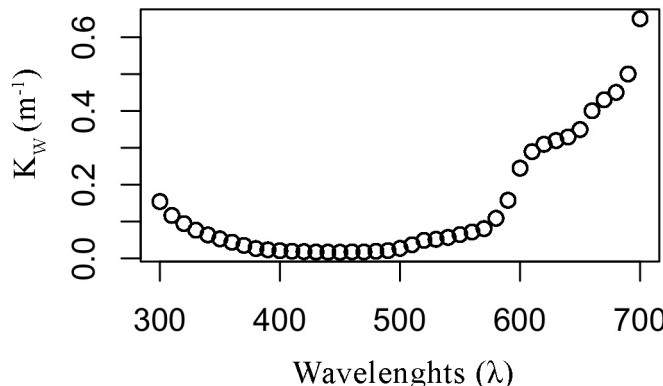

**Figure B1.** Values of the parameter $K_w$ as function of wavelengths between 300 to 700 nm.

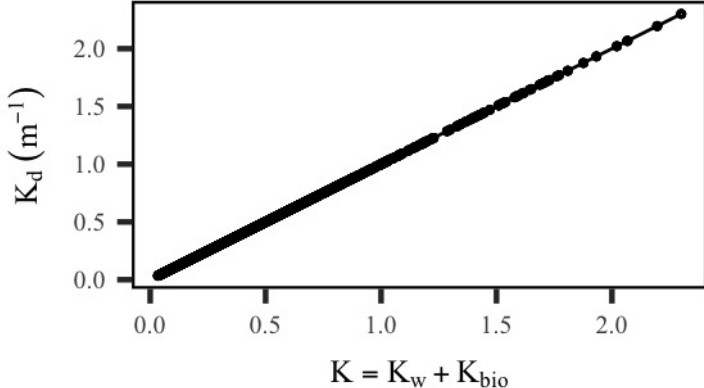

**Figure B2.** Correlation between values of field $K_d(\lambda)$ and modelled $K_d(\lambda)$ based on the absorption and scattering coefficients of pure seawater.



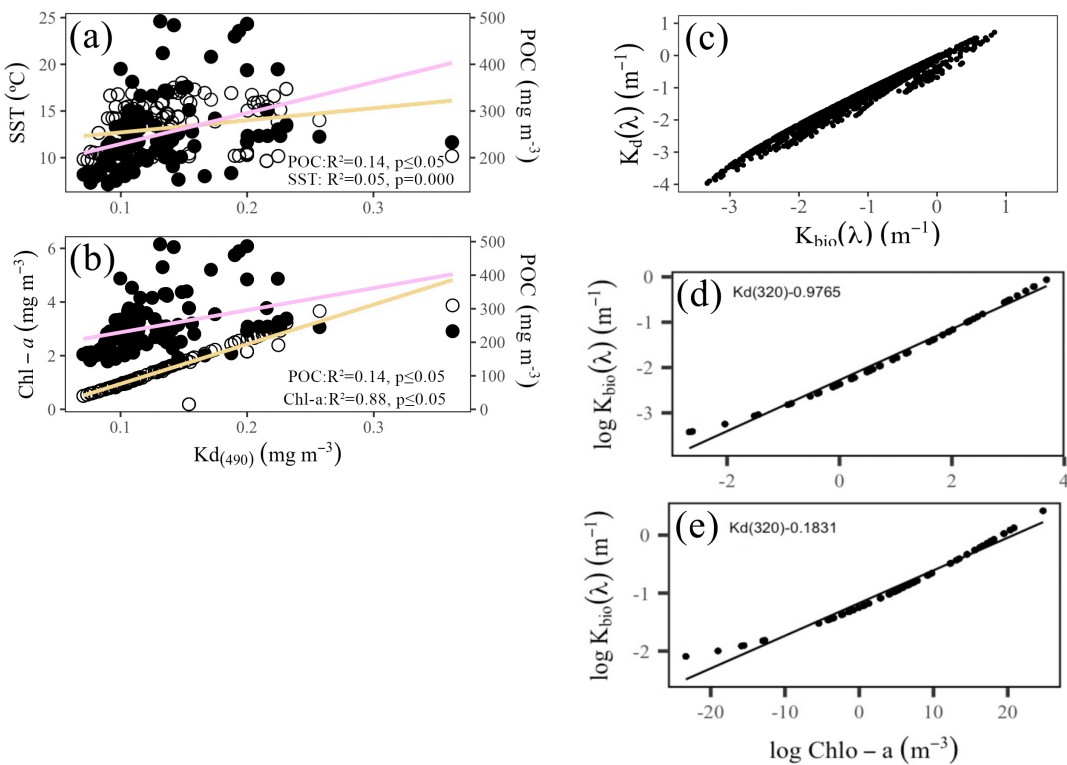

**Figure C1.** Correlation between remote sensing data (a,b). Correlation between calculated values of $K_{bio}(\lambda)$ and field obtained values of $K_d(\lambda)$ (c). Plots showing the log-log correlation between calculated values of Chl a and $K_{bio}(\lambda)$ at a high and low values of $K_d(320)$ (d,e).



**Figure C2.** Remote sensing data of particulate organic carbon (POC) for the months of summer and winter (upper and lower plots), and monthly average values for the transect illustrated with the black dashed line (middle section plots). Red dots represent the position of each sampling stations used to collect the data.



**Figure C3.** Remote sensing data of surface Chl a for the months of summer and winter (upper and lower plots), and monthly average values of Chl a for the transect illustrated with the black dashed line (middle section plots). Red dots represent the position of each sampling stations used to collect the data.

**Figure C4.** Remote sensing data of sea surface temperature (SST) for the months of summer and winter (upper and lower plots), and monthly average values for the transect illustrated with the black dashed line (middle section plots). Red dots represent the position of each sampling stations used to collect the data.


**Figure C5.** Remote sensing data of the attenuation coefficient of PAR light $k_d(490)$ for the months of summer and winter (upper and lower plots), and monthly average values for the transect illustrated with the black dashed line (middle section plots). Red dots represent the position of each sampling stations used to collect the data.




**Table A1.** Results of complete and decomposed time series analysis for the different atmospherical components influencing the amount of solar radiation penetrations the water column.

| Type | Measure | Temperature | UVR | Zu | Zv | Cloud Index |
|------|---------|-------------|-----|-----|-----|-------------|
| Complete | Frequency | 0.149 | 0.149 | 0.06 | 0.09 | 0.049 |
| | Trend | 0.76 | 0.317 | 0.235 | 0.166 | 0.549 |
| | Seasonality | 0.034 | 0.024 | 0.001 | 0.001 | 0.0003 |
| | Auto-correlation | 0.956 | 0.743 | 0.003 | 0.009 | 0.431 |
| | non-Linearity | 0.735 | 0.979 | 0.998 | 0.999 | 1 |
| | Skewness | 0.021 | 0.424 | 0.281 | 0.407 | 0.066 |
| | Kurtosis | 0.1 | 0.26 | 1 | 1 | 0.001 |
| | Hurts exponent | 1 | 0.999 | 0.542 | 0.575 | 0.887 |
| Decomposed | Auto-correlation | 0.877 | 0.909 | 0.074 | 0.049 | 0.062 |
| | non-Linearity | 0.999 | 1 | 0.954 | 0.667 | 0.777 |
| | Skewness | 0.121 | 0.363 | 0.006 | 0.064 | 0.243 |
| | Kurtosis | 0.226 | 0.277 | 1 | 1 | 0.221 |
| | Lyapunov exponent | 0.56 | 0.563 | 0.645 | 0.592 | 0.512 |





**Table B1.** Values of the the spectral absorption coefficients of pure sea water (aw, and values of the molecular scattering coefficient of pure sea water $(b_w)$, as determined by Smith and Baker, (1998).

| $\lambda$ (nm) | $a_w$ $(m^{-1})$ | $b_w$ $(m^{-1})$ | $\lambda$ (nm) | $a_w$ $(m^{-1})$ | $b_w$ $(m^{-1})$ |
|---|---|---|---|---|---|
| 300 | 0.1410 | 0.0262 | 510 | 0.0357 | 0.0026 |
| 310 | 0.1050 | 0.0229 | 520 | 0.0477 | 0.0024 |
| 320 | 0.0844 | 0.0200 | 530 | 0.0507 | 0.0022 |
| 330 | 0.0678 | 0.0175 | 540 | 0.0558 | 0.0021 |
| 340 | 0.0561 | 0.0153 | 550 | 0.0638 | 0.0019 |
| 350 | 0.0464 | 0.0134 | 560 | 0.0708 | 0.0018 |
| 360 | 0.0379 | 0.0120 | 570 | 0.0799 | 0.0017 |
| 370 | 0.0300 | 0.0106 | 580 | 0.1080 | 0.0016 |
| 380 | 0.0220 | 0.0094 | 590 | 0.1570 | 0.0015 |
| 390 | 0.0191 | 0.0084 | 600 | 0.2440 | 0.0014 |
| 400 | 0.0171 | 0.0076 | 610 | 0.2890 | 0.0013 |
| 410 | 0.0162 | 0.0068 | 620 | 0.3090 | 0.0012 |
| 420 | 0.0153 | 0.0061 | 630 | 0.3190 | 0.0011 |
| 430 | 0.0144 | 0.0055 | 640 | 0.3290 | 0.0010 |
| 440 | 0.0145 | 0.0049 | 650 | 0.3490 | 0.0010 |
| 450 | 0.0145 | 0.0045 | 660 | 0.4000 | 0.0008 |
| 460 | 0.0156 | 0.0041 | 670 | 0.4300 | 0.0008 |
| 470 | 0.0156 | 0.0037 | 680 | 0.4500 | 0.0007 |
| 480 | 0.0176 | 0.0034 | 690 | 0.5000 | 0.0007 |
| 490 | 0.0196 | 0.0031 | 700 | 0.6500 | 0.0007 |
| 500 | 0.0257 | 0.0029 | | | |



**Table B2.** Values of the coefficients $\chi(\lambda)$ and $e(\lambda)$ for pure sea water calculated by Morel (1988). (*) represent calculated values from fitting a polynomial equation.

| $\lambda$(nm) | $\chi(\lambda)$ | $e(\lambda)$ | $\lambda$ | $\chi(\lambda)$ | $e(\lambda)$ |
|---|---|---|---|---|---|
| *300 | 0.223 | 0.083 | 510 | 0.059 | 0.686 |
| *310 | 0.207 | 0.082 | 520 | 0.053 | 0.680 |
| *320 | 0.192 | 0.081 | 530 | 0.048 | 0.672 |
| *330 | 0.179 | 0.080 | 540 | 0.044 | 0.662 |
| *340 | 0.166 | 0.798 | 550 | 0.041 | 0.649 |
| *350 | 0.153 | 0.778 | 560 | 0.039 | 0.640 |
| *360 | 0.144 | 0.756 | 570 | 0.036 | 0.623 |
| *370 | 0.136 | 0.720 | 580 | 0.033 | 0.610 |
| *380 | 0.127 | 0.685 | 590 | 0.033 | 0.618 |
| *390 | 0.119 | 0.670 | 600 | 0.034 | 0.626 |
| 400 | 0.117 | 0.644 | 610 | 0.036 | 0.634 |
| 410 | 0.123 | 0.652 | 620 | 0.039 | 0.642 |
| 420 | 0.123 | 0.659 | 630 | 0.042 | 0.653 |
| 430 | 0.118 | 0.666 | 640 | 0.044 | 0.663 |
| 440 | 0.110 | 0.672 | 650 | 0.045 | 0.672 |
| 450 | 0.102 | 0.677 | 660 | 0.048 | 0.682 |
| 460 | 0.094 | 0.681 | 670 | 0.052 | 0.695 |
| 470 | 0.087 | 0.685 | 680 | 0.051 | 0.693 |
| 480 | 0.079 | 0.688 | 690 | 0.039 | 0.640 |
| 490 | 0.072 | 0.690 | 700 | 0.03 | 0.60 |
| 500 | 0.066 | 0.689 | | | |





*Author contributions.* PL defined the overall research problem and performed model analysis. All co-authors discussed the analyses and contributed to the text.

*Competing interests.* We the authors of the present manuscript declare that no competing interest are present

*Acknowledgements.* We would like to thank Sean Heseltine (Otago University, NZ) for the support with field work and Alva Curtsdotter
(University of New England) for general feedback and advice during model development.



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



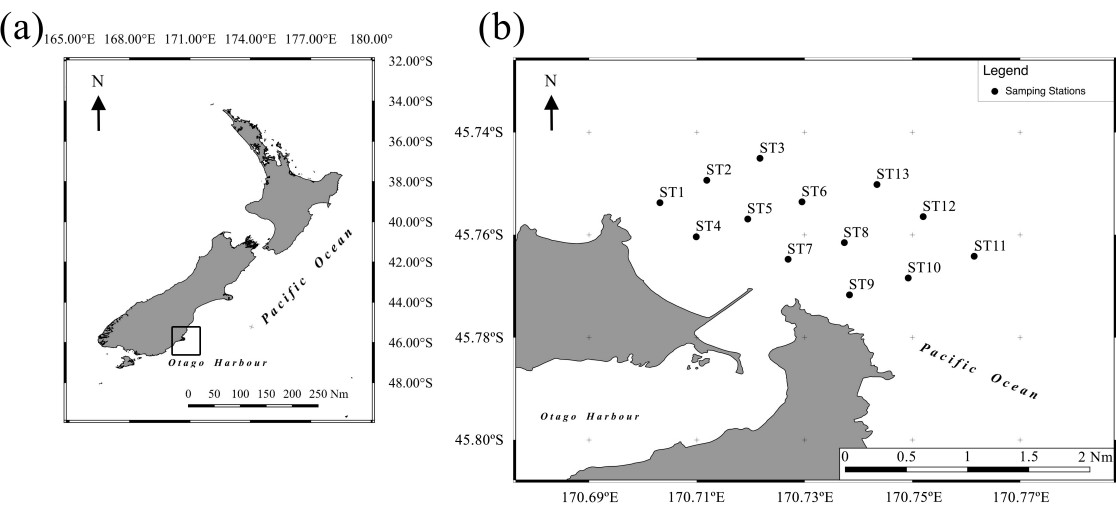

**Figure 1.** Grid of thirteen stations used for CTD casts and measurements of underwater solar radiation (a). Location of the Otago Harbour in the South island of New Zealand (b).



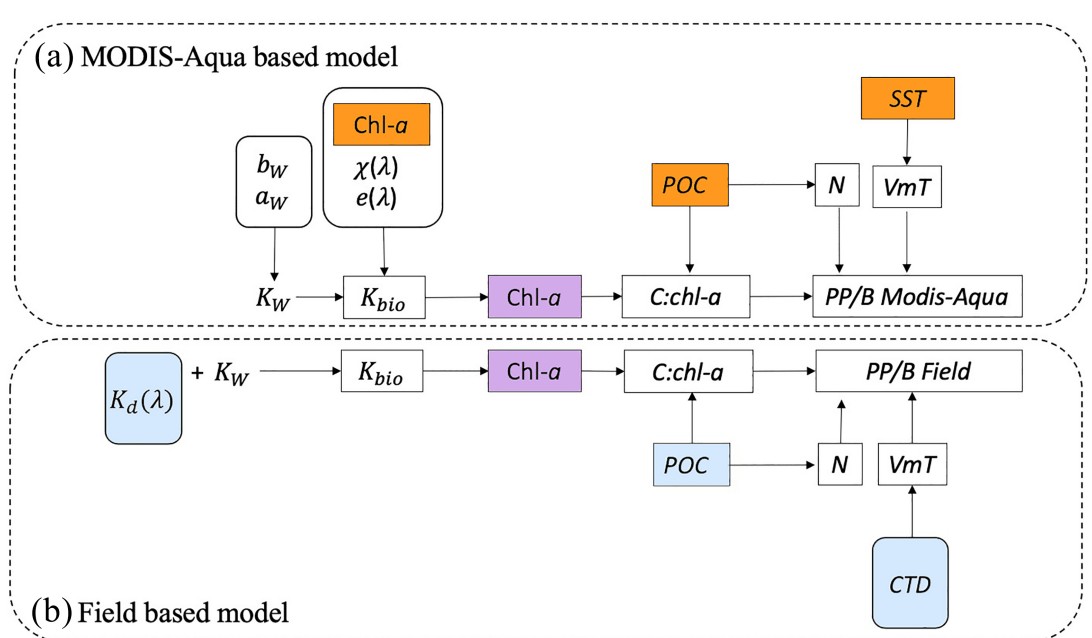

**Figure 2.** Schematic representation of the two model approaches used to predict the PP/B ratio. In the diagram yellow rectangles represent remote-sensing data and orange rectangles represent field obtained data.





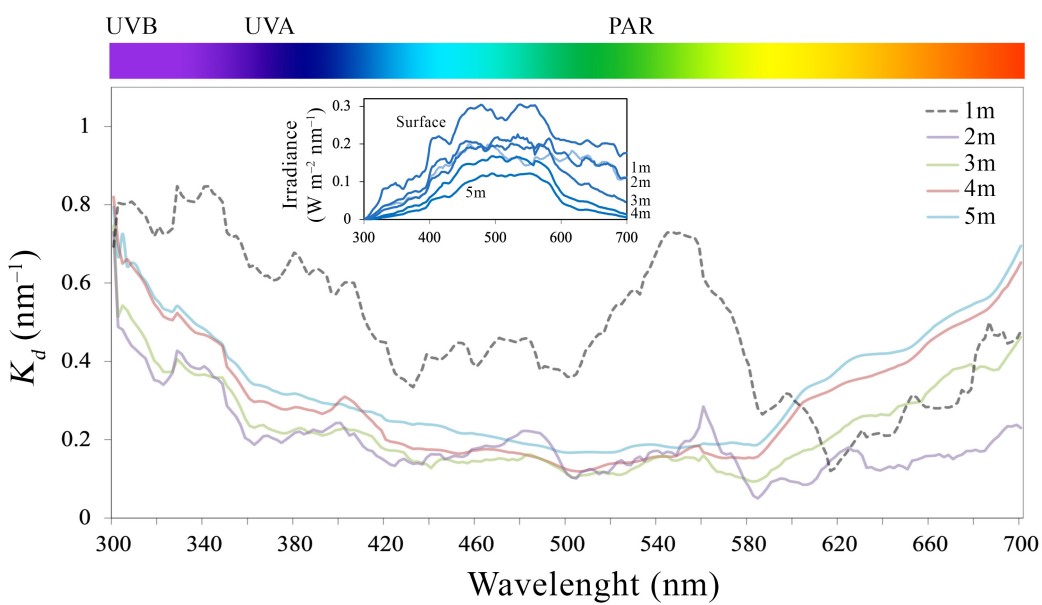

**Figure 3.** Profile of average values of the attenuation coefficient $(nm^{-1})$ at depths between 1 to 5 m across all sampling stations from 2016 to 2018 at the entrance of the Otago Harbour and irradiance values $(Wm^{-2}nm^{-1})$ for al wavelengths.





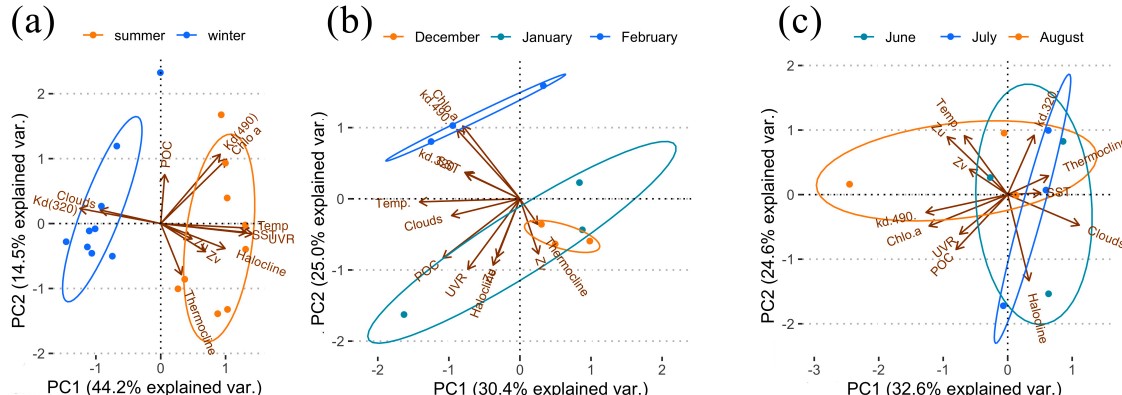

**Figure 4.** Principal component analysis (PCA) with average values of oceanographic and meteorological conditions during summer and winter (a). PCA analysis for each month during summer (b) and winter (c).




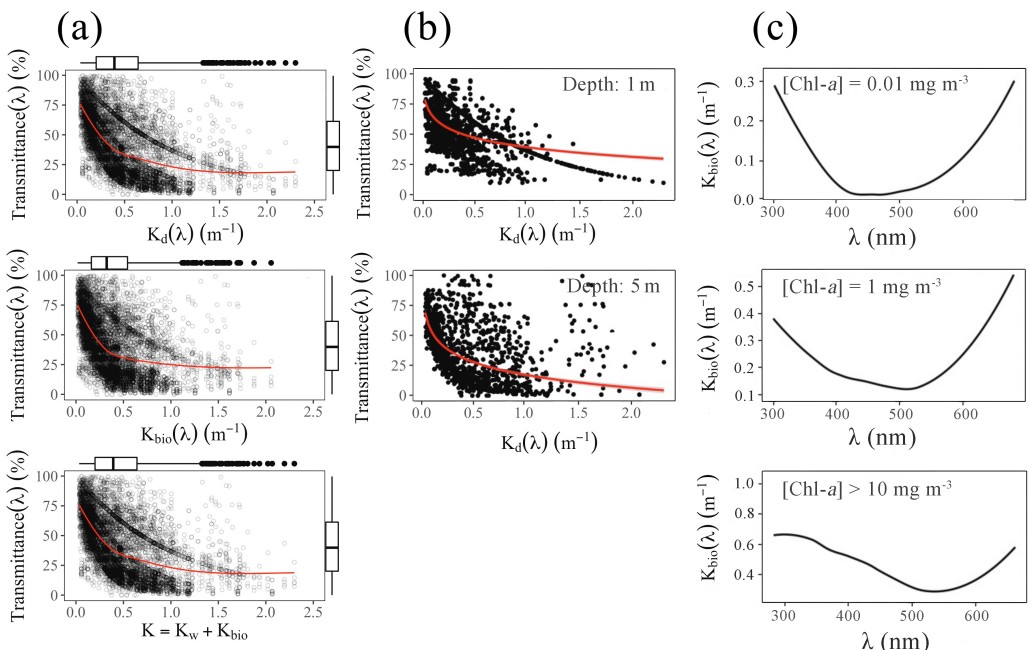

**Figure 5.** Correlation plots, with side box-plots, indicating position of the mean values and variability outside the upper and lower quartiles of the field-obtained transmittance $T(\lambda)$ against in-situ values of $K_d(\lambda)$, and for two modelled attenuation coefficient: $K_{bio}(\lambda)$ and $K(\lambda)$. Red lines indicated the adjusted log model (a). Correlation between field-obtained attenuation coefficients and transmittance at 1 m and 5 m depth (b). Changes in the attenuation coefficient $K_{bio}(\lambda)$ at three different concentrations of Chl a: 0.01, 1 and 10 $mg\,m^{-3}$ (c)





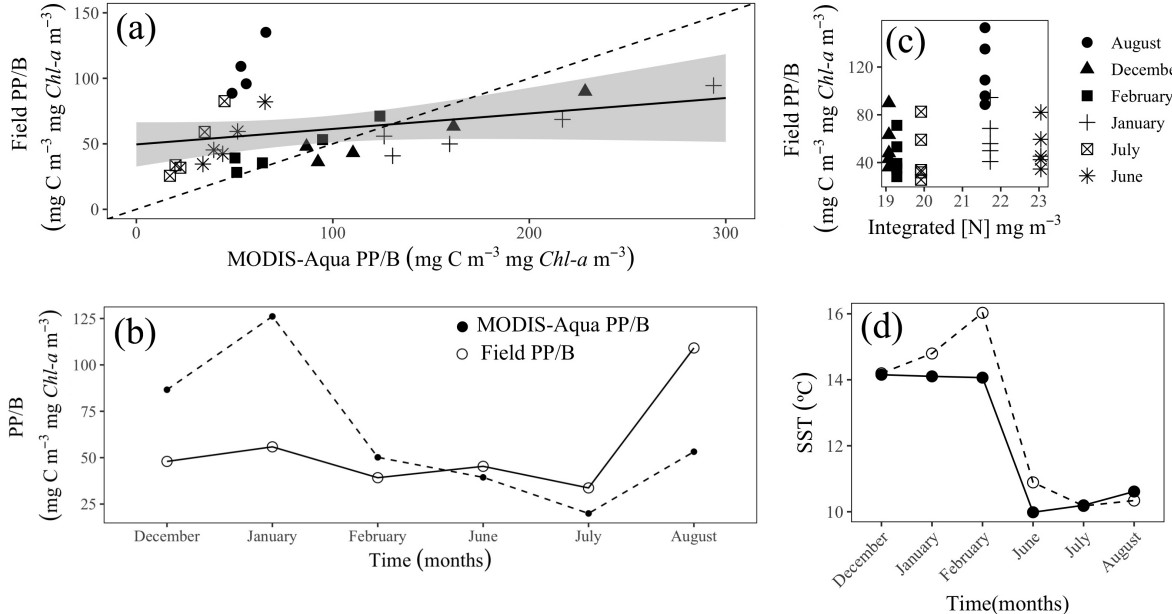

**Figure 6.** Correlation between the PP/B ratio obtained from remote sensing and field data using estimated values of Chl a (a). Seasonal changes in the PP/B ratio for both remote sensing and field obtained data-based models (b). Relationship between the integrated nitrogen values and the PP/B ratio from the field data base model (c). Seasonal trends in the sea surface temperature values from field obtained and remote sensing data-sets (d).





**Table 1.** Summary of oceanographic and meteorological variables used for PP/B model implementation and statistical analysis.

| Variables(units) | Abbrev. | Temporal Res. | Spatial Res. | Source |
|---|---|---|---|---|
| **Meteorological** | | | | |
| Wind speed ($km\ h^{-1}$) | $Zu, Zv$ | 1h | $20km$ | $*a, b$ |
| Temperature ($C$) | $Temp$ | 1h | $20km$ | $*a, b$ |
| Cloud cover (Scale 1 to 5) | $Cld$ | 1h | $1km^2$ | $*a, b$ |
| Global radiation ($Wm^{-2}$) | $UVR$ | 1h | $20km$ | $*a, b$ |
| **Oceanographic** | | | | |
| Chl a concentration($mg\ m^{-3}$) | $Chla$ | Monthly(3 years) | $1.5km^2$ | MODIS-aqua(NASA) |
| Sea surface temperature($C$) | $SST$ | Monthly(3 years) | $1.5km^2$ | MODIS-aqua(NASA) |
| Particular organic matter($mg\ m^{-3}$) | $POC$ | Monthly(3 years) | $1.5km^2$ | MODIS-aqua(NASA) |
| Attenuation coefficient PAR light($m^{-1}$) | $K_d(490)$ | Monthly(3 years) | $1.5km^2$ | MODIS-aqua(NASA) |
| Thermocline ($m$) | $Thr$ | Weekly(2 years) | $1km^2$ | CTD profiler |
| Pycnocline ($m$) | $Pyc$ | Weekly(2 years) | $1km^2$ | CTD profiler |
| Attenuation coefficient UV light($m^-1$) | $K_d(320)$ | Weekly(2 years) | $1km^2$ | Li-Cor LI1800UW |

*a = University of Otago, Department of Physic meteorological station. VAISALA HMP45A probe; Li-Cor LI200X Pyranometer; Vector A101M
Pulse Output Anemometer *b = New Zealand Meteorological service.



**Table 2.** List of symbol notation, units and constants values used for the model.

| Parameter | Description | Units |
|---|---|---|
| $K_d$ | Downwelling attenuation coefficient | $m^{-1}$ |
| $K$ | Theoretical attenuation coefficient | $m^{-1}$ |
| $K_{bio}$ | Biogenic attenuation coefficient | $m^{-1}$ |
| $a_W$ | Absoption spectra of pure sea water | $m^{-1}$ |
| $b_W$ | Scattering coefficient of pure sea water | $m^{-1}$ |
| $\chi$ | Derived function | dimensionless |
| $\alpha$ | Initial slope of production irradiance curve | $mgC(mgChla)^{-1}h^{-1}(\mu Em^{-1}s^{-1})$ |
| e | Derived function | dimensionless |
| Chl–a | Chlorophyll-a concentration | $mgm^{-3}$ |
| $K_W$ | Water decay constant | $m^{-1}$ |
| $K_n$ | Half saturation constant for nitrates | $mgm^{-3}$ |
| $T$ | Light transmittance | $(\%)m^{-1}$ |
| $I_Z$ | Light irradiance at depth | $Wm^{-2}$ |
| $z$ | Water column depth | m |
| N | Nitrogen yield | $mgm^3$ |
| $Vmt$ | Temperature correction function | dimensionless |
| $a$ | Maximum phytoplankton growth rate | $d^{-1}$ |
| $t$ | Ocean water temperature | °C |
| PP/B | Production-biomass ratio | $mgC(mgChla)^{-1}m^{-2}d^{-1}$ |





**Table 3.** Meteorological and oceanographic condition at the entrance of the Otago harbour, from 2016 to 2018.

|  |  | 2016 | | 2017 | | 2018 | |
|---|---|---|---|---|---|---|---|
|  |  | Summer | Winter | Summer | Winter | Summer | Winter |
| CTD temp(F) | X±Sd | 13.9±0.156 | 9.9±0.04 | 14.0±0.05 | 10.1±0.01 | 14.1±0.08 | 10.5±0.05 |
| °C | Max/Min | 13.7/14.2 | 9.8/9.9 | 13.9/14.1 | 10.1/10.1 | 13.9/14.2 | 10.4/10.6 |
| SST(M) | X±Sd | 14.8±1.64 | 10.9±0.57 | 14.2±0.91 | 10.3±0.07 | 16.0±0.90 | 10.2±0.28 |
| C | Max/Min | 13.2/16.9 | 10.2/11.6 | 13.4/15.5 | 10.3/10.4 | 10.3/10.4 | 9.85/10.5 |
| Thermocline(D) | X±Sd | 12.8±0.98 | 11.9±2.42 | 13.1±0.81 | 10.9±1.03 | 14.1±0.95 | 11.4±1.21 |
| m | Max/Min | 14.5/11.8 | 19.5/10.4 | 14.9/11.5 | 12.4/10.3 | 15.1/12.8 | 13.9/10.4 |
| Halocline(D) | X±Sd | 11.3±1.38 | 11.9±2.75 | 12.7±0.97 | 10.2±0.67 | 13.7±1.78 | 10.1±0.88 |
| m | Max/Min | 13.9/10.1 | 17.5/10.1 | 13.7/10.5 | 11.1/10.1 | 14.4/10.9 | 11.9/10.2 |
| Nitrate(M) | X±Sd | 21.7±5.61 | 23.0±2.91 | 19.1±1.89 | 19.9±4.38 | 19.3±1.87 | 21.6±5.0 |
| $mg\ m^{-2}$ | Max/Min | 15.9/29 | 20.7/27.0 | 16.6/21 | 16.1/25.9 | 16.8/21.1 | 16.8/28.2 |
| Chl a(D) | X±Sd | 3.12±2.18 | 1.61±1.84 | 2.98±1.24 | 2.14±1.21 | 3.33±2.53 | 2.61±2.11 |
| $mg\ m^{-3}$ | Max/Min | 4.48/0.11 | 3.98/0.03 | 5.54/0.08 | 3.51/1.11 | 4.63/0.02 | 7.78/2.89 |
| $K_d(490)$(M) | X±Sd | 0.21±0.09 | 0.09±0.04 | 0.30±0.01 | 0.14±0.03 | 0.33±0.07 | 0.11±0.05 |
| $m^{-1}$ | Max/Min | 0.14/0.38 | 0.05/0.16 | 0.29/0.32 | 0.11/0.20 | 0.30/0.39 | 0.06/0.189 |
| $K_d(320)$(F) | X±Sd | 0.38±0.01 | 0.309±0.05 | 0.37±0.02 | 0.238±0.02 | 0.32±0.02 | 0.313±0.05 |
| $m^{-1}$ | Max/Min | 0.39/0.36 | 0.36/0.23 | 0.39/0.34 | 0.25/0.21 | 0.33/0.31 | 0.36/0.24 |
| PAR(F) | X±Sd | 347.6±441 | 81.7±152 | 339±457 | 76.3±120 | 410±525 | 82.7±161 |
| $\mu mol\ m^{-2}s^{-1}$ | Max/Min | 1973/1 | 692/1 | 1871/1 | 715/0 | 1984/1 | 866/1 |
| Global UVR(F) | X±Sd | 1.26±1.07 | 0.37±0.48 | 1.26±1.12 | 0.39±0.53 | 1.28±1.14 | 0.33±0.43 |
| $Wm^{-2}$ | Max/Min | 4.15/0 | 2.08/0 | 4.26/0 | 2.47/0 | 4.09/0 | 1.98/0 |
| UVA(F) | X±Sd | 5.85±7.54 | 1.37±2.21 | 6.10±7.54 | 1.21±2.07 | 7.39±8.78 | 1.41±2.30 |
| $Wm^{-2}$ | Max/Min | 30.7/0 | 7.4/0 | 28.9/0 | 9.1/0 | 31.5/0 | 9.5/0 |
| UVB(F) | X±Sd | 0.487±0.66 | 0.07±0.10 | 0.494±0.54 | 0.06±0.09 | 0.63±0.8 | 0.07±0.11 |
| $Wm^{-2}$ | Max/Min | 2.77/0.01 | 0.49/0.01 | 2.57/0.01 | 0.49/0.01 | 2.93/0.01 | 0.51/0.01 |
| $Z_v$(D) | X±Sd | 3.89±108 | 2.64±86.7 | 5.20±105 | 0.58±63.2 | 5.80±82.3 | 0.77±56.3 |
| $m^2s^{-2}$ | Max/Min | 1093/–631 | 599/–513 | 851/–816 | 636/–501 | 500/–385 | 545/–428 |
| Cloud cover(D) | X±Sd | 2.66± | 3.08± | 2.67± | 2.92± | 2.71± | 2.66± |
| Idx | Max/Min | 1/5 | 1/5 | 1/5 | 1/5 | 1/5 | 1/5 |
| Temperature(F) | X±Sd | 16.6±4.68 | 6.85±5.0 | 16.3±4.64 | 6.91±4.31 | 17.5±4.88 | 6.60±4.34 |
| °C | Max/Min | 31.8/4.3 | 18/–6 | 29.3/2.5 | 20.2/–4.6 | 34.6/3.8 | 17.1/–6.3 |

(F) Average from field obtained measurements; (M) Average from the MODIS-Aqua sensor, (D) Average value derived from field measurements.