# Peer review of "Modelling the influence of light on the biological characteristics of coastal waters"

_Ocean Science, 2021_

## Referee Comment (RC2)

**Review of the paper entitled: "*Modelling the influence of light on the biological characteristics of coastal waters*" by P.F. Lagos et al. for publication in Ocean Science.**

**General comments**

This paper investigates the seasonal relationships between meteorological and oceanographic parameters over a three-year period and the effect light on the marine productivity (production/biomass ratio) in the Otago coastal area (New Zealand) through modelling approach. Although this study brings some interesting elements, it suffers from weaknesses both in terms of content and form.

Firstly, it seems to me that the authors completely obscure essential variables in their approach to determining the attenuation of solar radiation and its impact on marine productivity: 1) dissolved organic matter (DOM), and more particularly chromophoric DOM (CDOM), which is the main attenuator of solar UV radiation and which also very significantly absorbs visible radiation in aquatic ecosystems, furthermore in the coastal environment subject to riverine inputs. 2) The mixed layer depth (Zm), which strongly influences the occurrence of phytoplankton blooms and controls the impact of solar radiation.

Secondly, the study is not very extensive from a spatial and temporal point of view, which limits its scope and conclusions. The work was carried out for the Otago coastal waters but what is the implication for all the coastal areas (i.e., Case 2 waters)?

Then, the title of the paper is "*the influence of light on the biological characteristics".* However, the specific influence/contribution of light on/in the production-biomass ratio (PP/B), relative to the other parameters (POC, nitrogen, Chla...) is not at all assessed in this paper. There is no sensitivity study applied on light parameters (Kd, irradiances at specific depth…) to really determine their impacts on the PP/B ratios. More generally, there is a lack of cohesion and clarity between the objectives, the methodology, the results presented and the discussion which makes reading and understanding the manuscript quite difficult.

Also, as mentioned by the authors, their PP/B model is not validated by PP/B real measurements in the study area. In addition, it seems that there are no Chla concentrations measured on real water samples either to compare with the Chla data derived from bio-optical properties. This makes the results and conclusions much less impactful.

Finally, there are many syntax and editing errors (I cannot list all of them) and something wrong with the numbering of tables and figures.

For all these reasons, I recommend that this paper in the present form, which needs significant improvements, be rejected. Find below other comments/corrections.

**Other comments/corrections**

- Introduction, page 1, lines 18-20: "In the open ocean, solar radiation is the most important factor forcing atmospheric and ocean circulation…" What about wind?

- Introduction, page 1, lines 20-24: "*However, …..in coastal waters*". This sentence is not clear. Indeed, it is mentioned that changes in productivity depend on the complexity of coastal ecosystems which itself depends on factors including productivity.

- Introduction, page 2, line 34: Why the wavelength range 280-490 nm for Kd here?

- Everywhere in the manuscript: photosynthetically available radiation and ultraviolet radiation should be defined respectively as PAR and UVR the first time they are used with their associated wavelength range. By the way, "*PAR light*" and "*UVR light*" (or "*UV light*") should be replaced by "*PAR*" and "*UVR*". In addition, the interest of studying UVR should be better justified in the introduction.

- Everywhere in the manuscript: check the numbering of Tables and Figures. For instance, it seems that Table B2 is cited before Table B1, Figure 4 before Figure 3,…

- Everywhere in the manuscript: Sometimes the references are not cited in the chronological order (example line 38).

- Introduction, page 2, line 42: The reference *Gregg and Rousseaux, 2017* should be also cited here (Gregg WW, Rousseaux CS, 2017. Simulating PACE Global Ocean Radiances. Front. Mar. Sci. 4:60. doi: 10.3389/fmars.2017.00060).

- Introduction, page 2, line 42: Remove the reference "*Cao et al., 2014*" (cited twice).
- Material and Methods, page 3. Study site description (2.1): Figure 1 should be cited first in this section.

- Material and Methods, page 3, line 85: "*3.57 nm$^2$*". Something wrong with the unit I guess.

- Material and Methods, page 4, line 92: "*Table S1*". I cannot see Table S1.

- Material and Methods, page 4, equation 2 (Kd): I think there is a mistake, this is not λ1 and λ2 but Z1 and Z2. Moreover, in the denominator, this is not Z1 – Z2 but Z2 – Z1?

- Material and Methods, page 4. Kd calculation. It is not clear whether irradiance measurements have been made just above the sea surface (Es or Ed0+) 1) to correct in-water measurements in case of variation of surface irradiance due to cloud, and 2) to derive irradiance just below the sea surface (Ed0-). To determine Kd, sometimes it is much more relevant to use Ed0- derived from Ed0+ than the real measured Ed0- which can be affected by lens effects due to waves.

- Material and Methods, page 6, line 169: It is mentioned that Kw data from Smith and Baker (1981) were used. However, in Table B1 it is mentioned Smith and Baker (1998). In addition, why not using the Kw data from Morel et al. (2007)?

- Material and Methods, page 6, equation 8: "*Kbio = Kw = …*". I guess Kbio = Kd – Kw

- Material and Methods, page 6, line 172: "*In this study we assume that Kd(λ)…*" Here I think this is Kd(490) which is used to compute Chla concentration?

- Material and Methods, page 7, line 181: "*5mt depth*". Please correct.

- Results, page 7, line 204: "*Maximum surface levels of solar radiation between…*" Those data are issued the weather station measurements, right?

- Results, page 8, line 206: Replace "*peak of solar radiations*" by "*peak of irradiances*".

- Results, page 8, line 207: "*NCD*" is not defined I think.

- Results, page 8, line 216: Please provide wind speed in the same unit through the manuscript (kt, m s$^{-1}$ or km h$^{-1}$).

- Results, page 8, line 224 and elsewhere: Salinity is given in PSU but should be given without any unit.

- Results, page 8, line 226: "Kd(320)". This wavelength of 320 nm (at the limit between UVB and UVA) is not justified.

- Results, page 9, line 240: Remove the part: "*spectral values of*".

- Results, page 9, lines 257-258: "*The parameters used for the calculations of Chl a, aW and bW, were those of light as a function of each wavelength in pure sea water and led to some overestimated values of Chl a.*" This sentence is not clear.

- Results, page 10, line 283: "PP/B ratios from MODIS-Aqua and field obtained models had correlations values above~ 45". What does this number (45) mean? A coefficient of correlation of 0.45?

- Conclusion and discussion, page 10, lines 287-288: "*The atmospheric components included in this study only represent a sub-set of environmental mechanisms that control incident UVR reaching the surface of the water*". Why only UVR here and not PAR? The focus on UVR is not enough explicit in the introduction/objectives.

- Conclusion and discussion, page 10, line 300: "*UVA*" and "*UVB*" should be defined before.

- Conclusion and discussion, page 11, line 308: "*the correlation between wind and solar radiation became less clear (5).*" What does this number (5) mean?

- Conclusion and discussion, page 11, line 331: "*We found an strong*" Please correct.

- Conclusion and discussion, page 12, lines 338-339: "*The Otago coast is influenced by a high discharge of run – off from the Clutha river, bringing more particular organic matter into the study region, potentially driven by an increase of rainfall*". Here, DOM/CDOM should be mentioned.

- Conclusion and discussion, page 12, lines 341-343: "*These differences in rainfall plus higher solar radiation levels driven by ozone movement in the higher atmospheric levels due to wind speed have the potential to increase the productivity at a local scale, thus*

*maintaining higher levels of particulate matter in the first meters of the water column increasing Kd(λ) further*". This sentence is not clear at all. Please rewrite.

- Conclusion and discussion, page 12, line 352: "*We found spatial differences*".

- Conclusion and discussion, page 12, lines 355-356: "*We found that remote sensing attenuation coefficients Kd(490) correlated with in situ measurements of Kd(320)*". And not with in situ measurements of Kd(490)?

- Conclusion and discussion, page 12, lines 356-357: "*However, the process of obtaining valid measurements of in situ Kd(λ) was extensive and required comprehensive sampling to avoid erroneous measurements.*" I do not understand, why it is more difficult to obtain valid in situ Kd than modelled Kd?

- Conclusion and discussion, page 12, lines 364-356: "*Murdoch et al. (1990) it was expected to observe*". Please rewrite.

- Conclusion and discussion, page 13, lines 383: Replace "*lineal*" by "*linear*".

- Conclusion and discussion, page 13, lines 384-387: Actually it is not clear what it is compared in terms of Kd: Kd(490) versus Kd(Bio), measured versus modelled, Kd(Bio) from Chla versus Kd(Bio) from Kw.

- Conclusion and discussion, page 14, lines 420-421: "*Finally, the model implemented in this study was capable of interpreting seasonal changes in the PP/B ratio*". But it is not mentioned what is the importance of Kd in these seasonal changes.

---

## Author Comment (AC2)

Reviewer #2

**General comments**
This paper investigates the seasonal relationships between meteorological and oceanographic parameters over a three-year period and the effect light on the marine productivity (production/biomass ratio) in the Otago coastal area (New Zealand) through modelling approach. Although this study brings some interesting elements, it suffers from weaknesses both in terms of content and form.
Firstly, it seems to me that the authors completely obscure essential variables in their approach to determining the attenuation of solar radiation and its impact on marine productivity: 1) dissolved organic matter (DOM), and more particularly chromophoric DOM (CDOM), which is the main attenuator of solar UV radiation and which also very significantly absorbs visible radiation in aquatic ecosystems, furthermore in the coastal environment subject to riverine inputs. 2) The mixed layer depth (Zm), which strongly influences the occurrence of phytoplankton blooms and controls the impact of solar radiation.
Secondly, the study is not very extensive from a spatial and temporal point of view, which limits its scope and conclusions. The work was carried out for the Otago coastal waters but what is the implication for all the coastal areas (i.e., Case 2 waters)?
Then, the title of the paper is "the influence of light on the biological characteristics". However, the specific influence/contribution of light on/in the production-biomass ratio (PP/B), relative to the other parameters (POC, nitrogen, Chla...) is not at all assessed in this paper. There is no sensitivity study applied on light parameters (Kd, irradiances at specific depth…) to really determine their impacts on the PP/B ratios. More generally, there is a lack of cohesion and clarity between the objectives, the methodology, the results presented and the discussion which makes reading and understanding the manuscript quite difficult.
Also, as mentioned by the authors, their PP/B model is not validated by PP/B real measurements in the study area. In addition, it seems that there are no Chla concentrations measured on real water samples either to compare with the Chla data derived from bio-optical properties. This makes the results and conclusions much less impactful.
Finally, there are many syntax and editing errors (I cannot list all of them) and something wrong with the numbering of tables and figures.
For all these reasons, I recommend that this paper in the present form, which needs significant improvements, be rejected. Find below other comments/corrections.
Other comments/corrections

**Reply to general comments reviewer #2:**

We would like to thank the reviewer for the time spent reviewing this manuscript and for all the constructive critique. We acknowledge many of the weaknesses pointed out by the reviewer and the oversight on our part in the way the equations were added to the manuscript. However, these are only typos and they do not compromise the quality of the results presented in the manuscript. Additionally, we believe that, although it was beyond the scope of this study to measure all the variables that affect the attenuation coefficient, such as the DOM/CDOM, our study is a valuable contribution as most of the information presented here has never been published for the area of study. Similarly, we agree that the study would benefit if we could have added results and data to help infer conclusion to larger special scales, but we believe that the small spatial scale approach we used do not make the study less interesting or diminish its scientific value. As mentioned already there is not much published oceanographic information for the area and we believe that to address larger scale or more complicated issues one must start from the basic. Lastly, the improvements suggested by the reviewer can be easily

incorporated into the manuscript and we again apologize for any oversight in the presentation of the manuscript that make the reviewer doubt the validity of the results presented here. This study was written early in my PhD and vast improvements have been made since then.

**Reply to specific comments:**

- Introduction:
 page 1, lines 18-20: "In the open ocean, solar radiation is the most important factor forcing atmospheric and ocean circulation…" What about wind?
Answer: can be easily added

- Introduction, page 1, lines 20-24: "However, …..in coastal waters". This sentence is not clear. Indeed, it is mentioned that changes in productivity depend on the complexity of coastal ecosystems which itself depends on factors including productivity.
Answer: This paragraph has been improved for clarification.

- Introduction, page 2, line 34: Why the wavelength range 280-490 nm for Kd here?
Answer: It was written like this to illustrate the range of wavelengths we were using. However, we realize it is confusing and it has been changed to $Kd(\lambda)$
- Everywhere in the manuscript: photosynthetically available radiation and ultraviolet radiation should be defined respectively as PAR and UVR the first time they are used with their associated wavelength range. By the way, "PAR light" and "UVR light" (or "UV light") should be replaced by "PAR" and "UVR". In addition, the interest of studying UVR should be better justified in the introduction.
Answer: Agree, this issue has been fixed and a better justification for UVR added into the MS.

- Everywhere in the manuscript: check the numbering of Tables and Figures. For instance, it seems that Table B2 is cited before Table B1, Figure 4 before Figure 3,…
Answer: We checked the figure and table numbers and now they follow the expected order.

- Everywhere in the manuscript: Sometimes the references are not cited in the chronological order (example line 38).
Answer: References have been double checked and now are in order.

- Introduction, page 2, line 42: The reference Gregg and Rousseaux, 2017 should be also cited here (Gregg WW, Rousseaux CS, 2017. Simulating PACE Global Ocean Radiances. Front. Mar. Sci. 4:60. doi: 10.3389/fmars.2017.00060).
Answer: Agree, this issue has been amended

- Introduction, page 2, line 42: Remove the reference "Cao et al., 2014" (cited twice).
Answer: We cannot find this issue. Lines 39 to 43 currently are:
"These measurements have also allowed the modelling of light penetration through the water column (Taylor et al., 1997; Kim et al., 2015; Bowman et al., 2018). Current satellite open-access products, however, do not include measurements of short ultraviolet wavelength, and the use of open access remote sensing products is only possible in the visible band spectrum. For this reason, the behaviour of UV wavelengths in the ocean has been, in most cases, interpolated and interpreted from direct reflectance products through the implementation of complex models. (Mobley, 2001; Pan and Zimmerman, 2010; Li et al., 2018)".

- Material and Methods, page 3. Study site description (2.1): Figure 1 should be cited first in this section.
Answer: Reference to figure 1 has been added.

- Material and Methods, page 3, line 85: "3.57 nm2". Something wrong with the unit I guess.
Answer: Units have been replaced to $km^2$

- Material and Methods, page 4, line 92: "Table S1". I cannot see Table S1.
Answer: Table S1 is part of the Supplement B.

- Material and Methods, page 4, equation 2 (Kd): I think there is a mistake, this is not $\lambda 1$ and $\lambda 2$ but Z1 and Z2. Moreover, in the denominator, this is not Z1 – Z2 but Z2 – Z1?
Answer: Yes agree, we had typos in several of the equations and $\lambda$ should have been depth, same for the denominator order. The order should have been Z2 – Z1 as you correctly mention. However, we double checked the code and all the equation are implemented correctly in the code. Somehow mistakes were made when we inserted the equations into the manuscript. We apologize for the oversight and thank the reviewer for catching this mistake.
- Material and Methods, page 4. Kd calculation. It is not clear whether irradiance measurements have been made just above the sea surface (Es or Ed0+) 1) to correct in-water measurements in case of variation of surface irradiance due to cloud, and 2) to derive irradiance just below the sea surface (Ed0-). To determine Kd, sometimes it is much more relevant to use Ed0- derived from Ed0+ than the real measured Ed0- which can be affected by lens effects due to waves.
Answer: Agree this need corrections. We use measurements above the surface to correct the water measurements and measurements just below the surface (0.5m) to calculate Kd's. We added this information to the manuscript to clarify how measurements were made. Hopefully now should be clear.

- Material and Methods, page 6, line 169: It is mentioned that Kw data from Smith and Baker (1981) were used. However, in Table B1 it is mentioned Smith and Baker (1998). In addition, why not using the Kw data from Morel et al. (2007)?
Answer: We used the values from Smith & Baker 1981. But we agree we could have used the more recent values from Morel et al., (2007). The choice was rather arbitrary, as we believe it would not have much difference to the results of the PP/B.

- Material and Methods, page 6, equation 8: "Kbio = Kw = …". I guess Kbio = Kd – Kw
Answer: Yes this is one of several typos we have in the equations, as mentioned above. the correct equation is Kbio = Kd – Kw.

- Material and Methods, page 6, line 172: "In this study we assume that Kd($\lambda$)…" Here I think this is Kd(490) which is used to compute Chla concentration?
Answer: Yes this is correct; we now added the wavelength to avoid misinterpretation of the calculations.

- Material and Methods, page 7, line 181: "5mt depth". Please correct.
Answer: Correction added.

- Results, page 7, line 204: "Maximum surface levels of solar radiation between…" Those data are issued the weather station measurements, right?

Answer: Yes, that is correct, those measurements come from the weather station.

- Results, page 8, line 206: Replace "peak of solar radiations" by "peak of irradiances".
Answer: Correction added.

- Results, page 8, line 207: "NCD" is not defined I think.
Answer: Agree, now it is defined in page 4 line 116.

- Results, page 8, line 216: Please provide wind speed in the same unit through the manuscript (kt, m s-1 or km h-1).
Answer: Noted, this has been corrected.

- Results, page 8, line 224 and elsewhere: Salinity is given in PSU but should be given without any unit.
Answer: Agree and noted, this has been corrected
- Results, page 8, line 226: "Kd(320)". This wavelength of 320 nm (at the limit between UVB and UVA) is not justified.
Answer: Agree, we now better justify in the methods how and why we use Kd(320).

- Results, page 9, line 240: Remove the part: "spectral values of".
Answer: Noted, this has been deleted. Thanks

- Results, page 9, lines 257-258: "The parameters used for the calculations of Chl a, aW and bW, were those of light as a function of each wavelength in pure sea water and led to some overestimated values of Chl a." This sentence is not clear.
Answer: Sentence has been re-written and now reads: "Using the parameters Aw and Bw to calculate Chl a led to small overestimations of Chl a values"

- Results, page 10, line 283: "PP/B ratios from MODIS-Aqua and field obtained models had correlations values above ≈45". What does this number (45) mean? A coefficient of correlation of 0.45?
Answer: Yes this is the coefficient of correlation for the PP/B values between the two models. This has been corrected.

- Conclusion and discussion, page 10, lines 287-288: "The atmospheric components included in this study only represent a sub-set of environmental mechanisms that control incident UVR reaching the surface of the water". Why only UVR here and not PAR? The focus on UVR is not enough explicit in the introduction/objectives.
Answer: Agree. This has been replaced with "Light"

- Conclusion and discussion, page 10, line 300: "UVA" and "UVB" should be defined before.
Answer: Agree, this is now defined in the methods section.

- Conclusion and discussion, page 11, line 308: "the correlation between wind and solar radiation became less clear (≈5)." What does this number (5) mean?
Answer: This is the correlation value between wind speeds and solar radiation, equal to 0.05. Corrections have been made.

- Conclusion and discussion, page 11, line 331: "We found an strong" Please correct.

Answer: Correction made. Sentence now reads: " We found a connection..."

- Conclusion and discussion, page 12, lines 338-339: "The Otago coast is influenced by a
high discharge of run – off from the Clutha river, bringing more particular organic
matter into the study region, potentially driven by an increase of rainfall". Here,
DOM/CDOM should be mentioned.
Answer: Agree this could be very useful for us and for the study in general to report values like
this. Sadly, we have not found any studies reporting values of DOM or CDOM for the area.
We know based on our own and other researchers observations that DOM/CDOM can be high
but we personally did not measure those parameters so we didn't want to mention or report or
discuss information that is not properly backed up. In the same way this was one of the
motivation behind this MS. In general people know the area well, but there is very little
publiahed information about the area and no other study have reported attenuation coefficient
within the region.

- Conclusion and discussion, page 12, lines 341-343: "These differences in rainfall plus
higher solar radiation levels driven by ozone movement in the higher atmospheric levels
due to wind speed have the potential to increase the productivity at a local scale, thus
maintaining higher levels of particulate matter in the first meters of the water column
increasing Kd($\lambda$) further". This sentence is not clear at all. Please rewrite.
Answer: Sentence has been rewritten: Now reads: Although, rain data was not included in this
study, rainfall on a 6-month average for the Otago region is higher in summer (391mm)
compared to winter (367 mm) (Macara, 2015). The above average rainfall during summer,
together with higher levels of solar radiation can increase chlorophyll concentration at a local
scale, also maintaining greater levels of particulate matter in the first meters of the water
column and therefore predominantly higher surface Kd($\lambda$)".

- Conclusion and discussion, page 12, line 352: "We found spatial differences".
Answer: Change has been made

- Conclusion and discussion, page 12, lines 355-356: "We found that remote sensing
attenuation coefficients Kd(490) correlated with in situ measurements of Kd(320)". And
not with in situ measurements of Kd(490)?
Answer: Yes they also correlate with in situ measurements. Sentence has been added.

- Conclusion and discussion, page 12, lines 356-357: "However, the process of obtaining
valid measurements of in situ Kd($\lambda$) was extensive and required comprehensive
sampling to avoid erroneous measurements." I do not understand, why it is more
difficult to obtain valid in situ Kd than modelled Kd?
Answer: The sentence was making reference to the inherent logistic difficulties of going
sampling every day and waiting for the proper environmental condition to get accurate
measurements versus downloading files from satellite measurements. Would like to point out
that this was not meant as a way of complaining.

- Conclusion and discussion, page 12, lines 364-356: "Murdoch et al. (1990) it was
expected to observe". Please rewrite.
Answer: Sentence has been rewritten, now it reads: it was expected to observe a less sharp
linear decay of light with depth and not the abrupt decay in the..."

- Conclusion and discussion, page 13, lines 383: Replace "lineal" by "linear".

Answer: Thanks, typo has been corrected.

- Conclusion and discussion, page 13, lines 384-387: Actually it is not clear what it is compared in terms of Kd: Kd(490) versus Kd(Bio), measured versus modelled, Kd(Bio) from Chla versus Kd(Bio) from Kw.
Answer: Agree this sentence can be further clarified.

- Conclusion and discussion, page 14, lines 420-421: "Finally, the model implemented in this study was capable of interpreting seasonal changes in the PP/B ratio". But it is not mentioned what is the importance of Kd in these seasonal changes.

Answer: Thanks for the feedback and we agree that this can be easily developed further in this section.